# State of the interactomes: an evaluation of molecular networks for generating biological insights

Sarah N Wright, Scott Colton, Leah V Schaffer, Rudolf T Pillich, Christopher Churas, Dexter Pratt & Trey Ideker

## Abstract

**Advancements in genomic and proteomic technologies have powered the creation of large gene and protein networks ("interactomes") for understanding biological systems. However, the proliferation of interactomes complicates the selection of networks for specific applications. Here, we present a comprehensive evaluation of 45 current human interactomes, encompassing protein-protein interactions as well as gene regulatory, signaling, colocalization, and genetic interaction networks. Our analysis shows that large composite networks such as HumanNet, STRING, and FunCoup are most effective for identifying disease genes, while smaller networks such as DIP, Reactome, and SIGNOR demonstrate stronger performance in interaction prediction. Our study provides a benchmark for interactomes across diverse biological applications and clarifies factors that influence network performance. Furthermore, our evaluation pipeline paves the way for continued assessment of emerging and updated interaction networks in the future.**

**Keywords** Gene Prioritization; Interaction Prediction; Interactome; Network; Systems Biology
**Subject Categories** Computational Biology; Proteomics

## Introduction

Molecular networks ("interactomes") are a crucial tool in biomedical research for translating complex biological data into actionable insights. These networks are constructed from diverse genomic, proteomic, biochemical, and statistical data sources to represent known or measured biological interactions among genes, proteins, and other biological entities (Tolani et al, 2021). Interactions encompass a wide range of types, including physical protein-protein interactions, regulatory relationships, signaling and metabolic pathways, and functional associations, each of which contribute to our understanding of molecular and cellular mechanisms.

By leveraging the information in these interactomes, network biology offers the potential to understand gene and protein functions at a systems level, thus enabling a comprehensive understanding of biological and disease processes. Network approaches have enabled the interpretation of genome-wide association studies (GWAS) (Leiserson et al, 2013; Wang et al, 2019; Visonà et al, 2024; Carlin et al, 2019), prioritized disease genes (Magger et al, 2012; Prajapati and Emerson, 2020), accelerated the discovery of gene functions (Kim and Lee, 2017; Depuydt and Vandepoele, 2021), and revealed functional similarities and differences between species and cell types (Wan et al, 2015; Huttlin et al, 2021). The success of each approach can depend critically on selecting an appropriate interactome for study. However, the rapid generation of biological data has led to a proliferation of available networks, making network selection increasingly challenging.

No matter the type of analysis performed, the ability of a network approach to generate biological insights depends on what information is and is not present in the interactome. It is widely recognized that our knowledge of biological interactions is incomplete, especially for experimentally supported interactions (Menche et al, 2015; Kovács et al, 2019; preprint: Brunson et al, 2023). However, the extent to which gaps are skewed towards certain genes or biological processes is less well understood. Many factors could potentially cause skew, including experimental constraints or biases towards highly studied or highly expressed genes (Gillis et al, 2014). Beyond gaps in interactome knowledge, network resources may contain false positive interactions, which can dilute biological signals. Understanding these biases and other effects is important, as any bias in an interactome will likely be reflected in analyses that use that interactome.

Several years ago, we established methods for systematically evaluating human molecular networks and demonstrated that large composite interactomes provided the best performance for prioritizing disease genes (Huang et al, 2018). This work culminated in the Parsimonious Composite Network (PCNet), a consensus network that includes the most supported interactions across different network resources while excluding potentially spurious relationships (Huang et al, 2018). Given the continued increases in size and quantity of interactomes, ongoing benchmarking of molecular networks is essential for guiding network selection. Accordingly, here we present an updated and expanded evaluation, providing the most extensive assessment to date of the

---

Department of Medicine, University of California San Diego, La Jolla, CA 92093, USA. ✉E-mail: tideker@ucsd.edu

contents and performance of publicly available human molecular networks. Concomitant with these efforts is the release of PCNet2.0, a new and improved consensus human gene network.

In addition to the established procedure for evaluating the performance of interactomes via disease gene prioritization, we introduce an interaction-centric evaluation metric. This approach leverages interaction prediction algorithms, which provide an efficient method for addressing network incompleteness and have previously been used to identify and prioritize novel interactions (Johnson et al, 2021; McDowall et al, 2009). Earlier benchmarking studies have assessed the performance of different prediction algorithms across a small number of human interactomes (Zahiri et al, 2020; Wang et al, 2023). Building on these studies, we now implement interaction prediction across a wide range of interactomes to assess the influence of the underlying network architecture on prediction accuracy. We define external interaction sets for this evaluation and create an in silico assessment pipeline for novel predictions using AlphaFold-Multimer (preprint: Evans et al, 2021). Through this comprehensive evaluation, we aim to provide an up-to-date survey of 45 interactomes, equipping researchers with the information necessary to navigate network selection and thereby power the generation of biological insights across various applications. Our consensus networks are easily accessible via the Network Data Exchange (NDEx (Pratt et al, 2017; Pillich et al, 2021), www.ndexbio.org), and our evaluation pipeline is also available (https://github.com/sarah-n-wright/Network_Evaluation_Tools), allowing for ongoing analysis of biological networks.

## Results

### A plethora of interaction networks from diverse but overlapping sources

We performed a census of current biomolecular network resources, focusing on gene and protein-centric human interactomes. Our survey identified 45 publicly available networks (Appendix Table S1), which we classified into three categories: Experimental —networks formed from a single experimental source, Curated— networks curated manually from literature sources, and Composite —networks incorporating multiple curated or experimental databases. Across these categories, we observed substantial diversity in the network features and data sources (Fig. 1A). For example, while 93% of the interactomes incorporated physical protein-protein interactions (PPIs), fewer than 25% contained information from genome or protein structural similarities. The majority of network resources we surveyed (71%) contained interaction evidence from multiple species. We excluded all non-human interactions from these networks for our human-centric analysis, except where the authors explicitly used orthologous interactions from model species to infer human networks.

All networks were standardized to map gene identifiers to NCBI Gene IDs, remove duplicates and self-interactions, and filter to human interactions where applicable. Composite networks tended to be much larger than experimental and curated networks, both in the number of genes represented and the number of reported interactions (Fig. 1B). Building on the 21 networks analyzed in our prior publication (Huang et al, 2018), we noted that 14 of these had

been significantly updated while 7 represented static databases; we also extended the corpus with 24 additional interactomes. Of the 14 updated networks, the Human Reference Interactome (Luck et al, 2020) (HuRI) grew the most, with a 2.6-fold increase in genes and a 4.6-fold increase in interactions compared to the previously evaluated Human Interactome (Rolland et al, 2014) (HI-II-14) (Fig. EV1A). While the number of distinct databases has increased substantially, most available networks rely on similar sources and have extensive dependencies (Fig. 1C). The interactomes most frequently identified as dependencies of other network resources were IntAct (Orchard et al, 2014) (20/45 databases), BioGRID (Stark et al, 2006) (19/45 databases), DIP (Xenarios et al, 2000) (16/45 databases), MINT (Licata et al, 2012) (15/45 databases), and HPRD (Peri et al, 2003; Mishra et al, 2006; Keshava Prasad et al, 2009) (15/45 databases).

### Gaps remain in coverage of the human proteome

By definition, a network analysis can only discover genes and processes that exist within the interactome selected for study. Despite 99% of protein-coding genes (as defined by the HUGO Gene Nomenclature Committee, HGNC (Seal et al, 2023)) being represented in at least one interactome, we found that their distribution varies widely across networks (Fig. 2A). Other genetic elements, such as non-coding RNAs and pseudogenes, are sparsely represented (Fig. 2B). Looking at the gene citation counts for protein-coding genes, we found that genes in PID v2.0 (Pillich et al, 2023), DIP (Xenarios et al, 2000), and PhosphoSitePlus (Hornbeck et al, 2015) showed the most skew towards high citation counts (Fig. 2C). Further, we observed a significant correlation between a gene's network coverage, which we defined as the number of interactomes a gene appears in, and the gene's citation count (Fig. EV1B, $r_{s,citation} = 0.80$, $p = 1 \times 10^{-50}$). The relationship between network coverage and citation was diminished but not eliminated when considering only experimental networks (Fig. EV1C, $r_s = 0.48$, $p = 1 \times 10^{-50}$). High mRNA expression (Data ref: GTEx Portal, 2017) and protein abundance (Uhlén et al, 2015, Data ref: The Human Protein Atlas, 2023) also significantly correlated with increased network coverage ($r_{s,mRNA} = 0.59$, $r_{s,protein} = 0.40$, $p = 1 \times 10^{-50}$, Fig. EV1D,E). Among the interactomes, we observed that experimental networks tended to show skew towards highly expressed genes and abundant proteins (Fig. 2C). These same networks also tended to enrich for highly conserved genes (Fig. EV1F) and demonstrated under-enrichment for tissue-specific genes (Appendix Figs. S1 and S2). After adjusting for mRNA expression and protein abundance, the correlation between network coverage and gene citation was partly reduced ($r_{s,citation} = 0.59$, $p = 1 \times 10^{-50}$), indicating that expression levels contribute to, but do not completely explain, the observed citation bias (Fig. EV2A).

We next hypothesized that gaps and biases in the genes represented in each interactome would cause gaps in knowledge of biological function. Gene set enrichment analysis using Gene Ontology (GO) annotations showed that experimental interactomes favor translational machinery while being systematically under-enriched for receptor-ligand and cyclase activities, possibly due to experimental limitations (Fig. 2D). Furthermore, many small curated and composite networks such as SIGNOR (Lo Surdo et al, 2022), PID v2.0 (Pillich et al, 2023), and DIP (Xenarios et al,

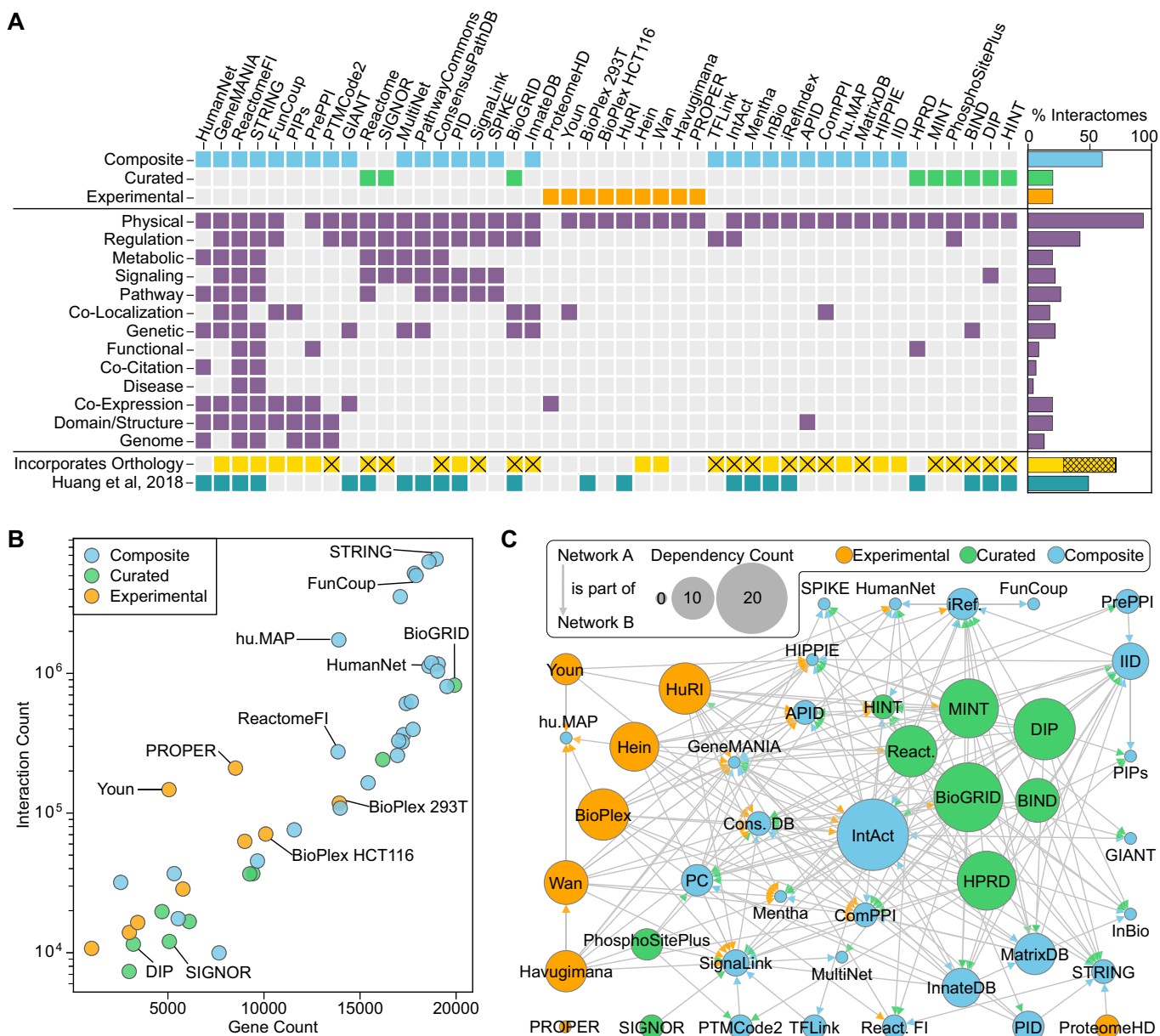

**Figure 1. Network contents, sources, and dependencies.**

(A) Interactome classification, interaction types, and network features. Crosses indicate that non-human interactions were filtered out in data processing. The bar chart shows the percentage of all interactomes with each feature. See Appendix Table S2 for interaction type definitions. (B) Interactome size plotted by the number of genes versus the number of interactions. Counts represent distinct genes and interactions after data processing and conversion of identifiers to NCBI Gene IDs. Color indicates interactome classification. (C) Dependency relationships between interactomes, colored by interactome classification. Arrows represent that the source interactome is incorporated into the target interactome. Node size indicates the "Dependency Count," defined as the number of times a given interactome was identified as a source by other evaluated network sources (Methods). The arrow color indicates the interactome classification of the source network. PC: Pathway Commons; iRef: iRefIndex; React: Reactome, Cons DB: ConsensusPathDB.

2000) were enriched for inflammation and immune response genes, while protein glycosylation and transporter activity were generally underrepresented.

Across all interactomes, 23.6 million distinct interactions were reported, with 98% representing interactions between proteins or protein-coding genes. The majority of interactions (73.5%) were unique to a single network, while those reported in 13 or more networks comprised only 1% of the interactions (Fig. 2E). High

expression and citation levels were generally associated with high interaction density among protein-coding genes (Fig. 2F). For example, among the 266 most cited genes, 80% of possible pairwise interactions were reported at least once, while only 0.1% of possible interactions were reported between the 480 genes with only one citation. However, an increased density of interactions was observed among a subset of genes with low mean mRNA expression - driven by interactions between tissue-specific genes of whole blood, thyroid, and

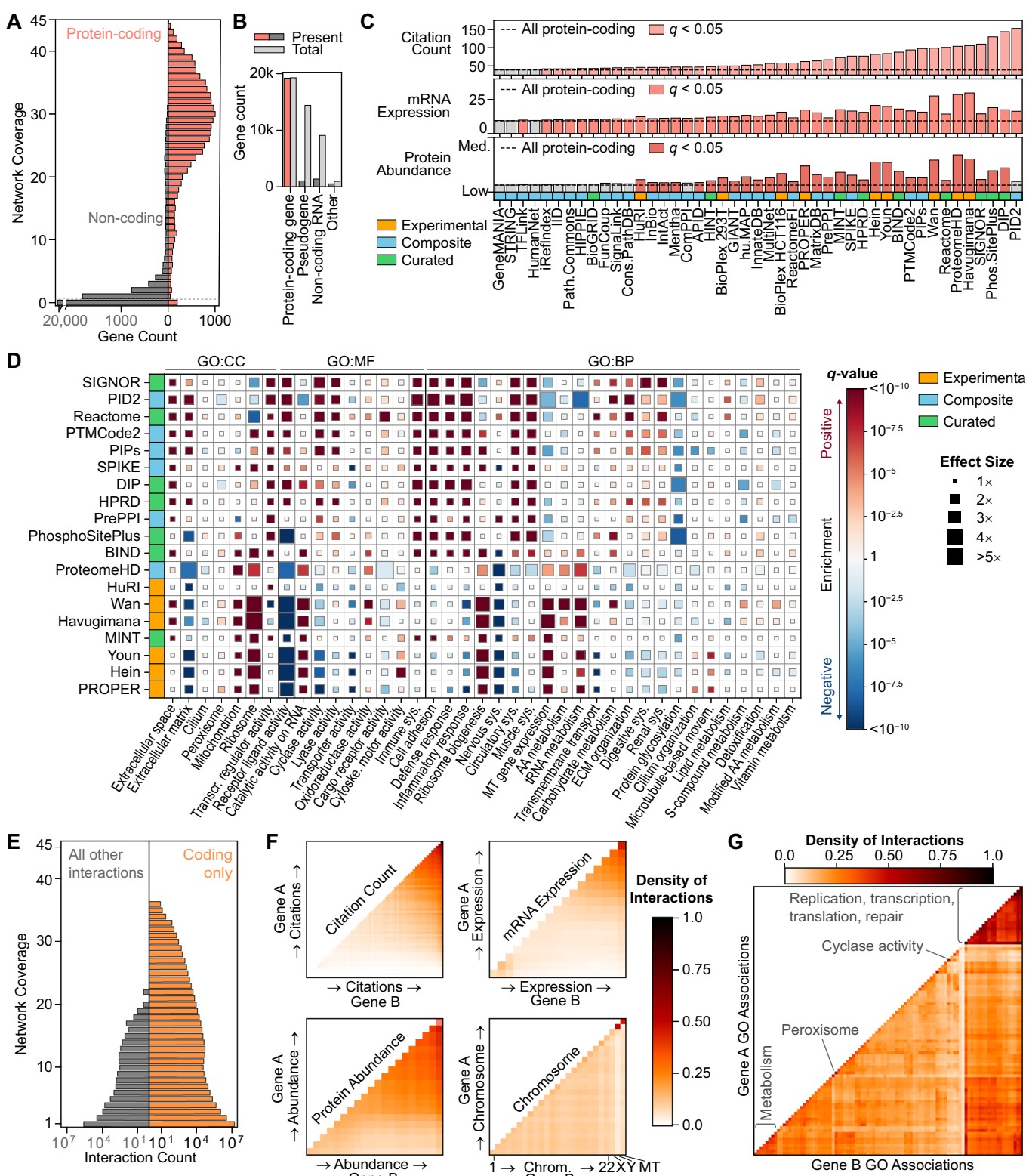

skin (Fig. EV2B). In contrast, the interaction density was largely uniform within and between chromosomes, except for the Y and mitochondrial chromosomes (Fig. 2F).

Analysis of interaction density among GO Biological Process annotations highlighted a high density of interactions among genes involved in a broad range of nucleic acid and metabolic functions (Fig. 2G). In contrast, genes associated with some functions and components, such as cyclase activity and the peroxisome, had a high density of interactions amongst themselves but fewer interactions with outside genes. To further test each interactome's

**Figure 2. Biological representation of interactome genes and interactions.**

(A) Network coverage of individual genes after mapping to NCBI Gene IDs, separated into protein-coding and non-coding genes based on HGNC locus type. (B) Presence of genes in at least one interactome by HGNC locus type, compared to total genes of each type. (C) Median citation count, mRNA expression, and protein abundance for protein-coding genes in each interactome. Dashed line represents the median of all protein-coding genes. Colored bars show networks with significantly higher median values than the baseline of all protein-coding genes (permutation test, $q < 0.05$, Bonferroni correction). (D) Functional enrichment of network genes across the Gene Slim Ontology (Fisher's Exact test, BH correction). Terms are grouped based on GO branches: BP: biological process, CC: cellular component, MF: molecular function. To highlight GO terms that are poorly represented in at least one interactome, the visualization includes networks with <10,000 genes and terms with at least one nominally significantly under-enriched network ($p < 0.01$). Full results are in Dataset EV1. (E) Network coverage of distinct interactions after interactome processing, separated into those between two protein-coding genes ("Coding only") and those involving at least one non-coding gene ("All other interactions"), as defined by HGNC locus type. (F, G) Interaction density of all distinct interactions as a function of gene annotations. Genes are binned based on increasing citation count, mRNA expression percentile, protein abundance percentile, and chromosome (F) or GO Slim annotations (G). The interaction density is calculated per pair of bins as the number of observed interactions divided by the number of possible interactions between all combinations of gene A and gene B for a bin pair. The diagonal entries, therefore, represent the interaction density between genes with similar annotation values, while off-diagonal entries represent the interaction density between genes with differing annotation values.

ability to represent GO annotations, we tested physiological gene function prediction using guilt-by-association (Ballouz et al, 2017). HumanNet (Kim et al, 2022), Wan (Wan et al, 2015), Havugimana (Havugimana et al, 2012), DIP (Xenarios et al, 2000), Reactome (Gillespie et al, 2022), and PTMCode2 (Minguez et al, 2015) showed the best recapitulation of GO annotations (Fig. EV2C), with performance generally best for cellular compartments (Fig. EV2D–F). However, it should be noted that some resources, such as HumanNet and Havugimana, utilize GO to prioritize network interactions during construction (Kim et al, 2022; Havugimana et al, 2012).

## Large composite networks remain the most powerful for disease gene prioritization

Next, we assessed each interactome's ability to recover a range of disease-associated gene sets using a network propagation framework (Methods, Fig. 3A). This approach is based on the principle that genes in close proximity within a biological network are likely to share biological functions. For disease gene set recovery, we used network propagation to rank all genes within an interactome based on their proximity to a known set of disease genes. We then assessed the position of held-out members of the gene set within these propagation ranks to determine how well the network structure recovered known disease associations. To quantify this gene set recovery performance, we compared the results to ranks generated with shuffled interactomes, yielding performance Z-scores (Huang et al, 2018). Our analysis evaluated gene set recovery performance for all interactomes using diverse gene sets sourced from DisGeNET (Piñero et al, 2020) ("Literature"; $n = 906$) and the GWAS catalog (Sollis et al, 2023, Data ref: GWAS Catalog, 2024) ("Genetic"; $n = 699$). Literature gene sets, being larger, generally showed better interactome coverage and gene set recovery performance (Fig. EV3A).

Averaging the performance Z-scores across all gene sets revealed that large composite networks HumanNet (Kim et al, 2022) and STRING (Szklarczyk et al, 2023) were most effective for prioritizing Literature and Genetic disease genes, respectively (Fig. 3B). Given the previously established relationship between network size and gene set recovery performance (Huang et al, 2018), we also computed size-adjusted performance metrics by regressing the number of interactions per network. Smaller networks, such as DIP (Xenarios et al, 2000), SIGNOR (Lo Surdo et al, 2022), and PTMCode2 (Minguez et al, 2015), rose in the rankings after adjustment, indicating that their interactions are

highly informative on a per-edge basis (Fig. 3B). When controlling for gene set coverage, a positive correlation was maintained between performance and network size (Fig. EV3B). However, the magnitude of this effect was significantly reduced ($p_{Literature} = 8.8 \times 10^{-34}$; $p_{Genetic} = 4.1 \times 10^{-6}$; Fig. EV3C), indicating that larger networks are, in part, advantaged by capturing a greater proportion of the known genes.

Interactome performance was consistent across Literature and Genetic gene sets ($r_s = 0.80$, $p = 3.0 \times 10^{-11}$), with exceptions for networks such as ConsensusPathDB (Kamburov et al, 2008), DIP (Xenarios et al, 2000), and ProteomeHD (Kustatscher et al, 2019), which showed discrepancies across gene set sources (Fig. 3C). These exceptions indicate that some interactomes are better suited to analyzing certain data types. To address potential circularity in our analysis due to literature curation—a concern that arises because some gene sets and interactomes may be derived from overlapping literature sources—we also assessed performance with additional gene sets derived from GWAS published after July 2023 ("Genetic 2023+"; $n = 48$) and experimental studies published after January 1, 2024 ("Experimental"; $n = 17$). Thus, the Genetic 2023+ and Experimental sets were generated from data available only after the release of any interactome in our corpus that utilized text mining of biomedical literature. Performance rankings for Genetic gene sets were highly correlated with those for Genetic 2023+ gene sets ($r_s = 0.84$, $p = 4.2 \times 10^{-13}$, Fig. 3C), indicating minimal bias from literature curation for Genetic gene sets. Correspondence to Experimental gene sets was also positively correlated, though some interactomes, such as ConsensusPathDB (Kamburov et al, 2008), showed reduced performance for non-literature gene sets. Overall, the strong correlations in network rankings across different gene sets underscored the robustness of our analysis against concerns of circularity due to literature curation.

## Not all interaction types are created equal

The highest-performing networks for gene set recovery were composite networks comprising many evidence types, leading us to hypothesize that different interaction types did not contribute equally to the observed performance. We examined the contributions of the various interaction types in two large network databases: HumanNet (Kim et al, 2022) and GeneMANIA (Mostafavi et al, 2008). We defined interaction-type-specific networks from each source, ranging from the small HumanNet orthology network to the large GeneMANIA co-expression network (Fig. EV3D). The two physical interaction networks were highly

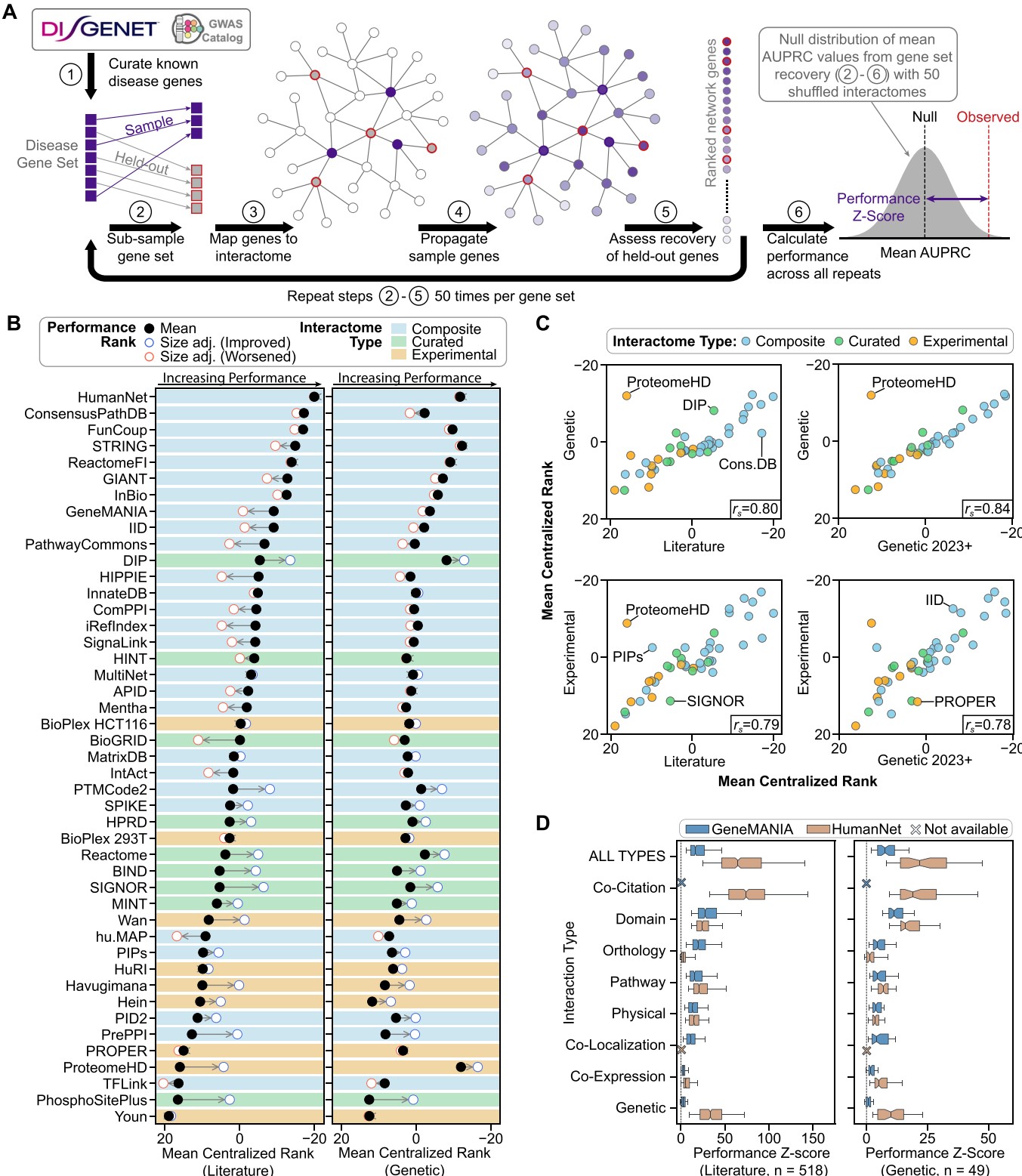

**Figure 3.  Disease gene set recovery performance.**

(A) Overview of the gene set recovery pipeline to generate performance Z-scores based on areas under the precision-recall curve (AUPRC). (B) The mean network performance of each interactome across all Literature (left) and Genetic (right) gene sets, measured as the mean centralized rank of interactomes. A more negative rank indicates better relative performance. Arrows and colored circles show the mean size-adjusted rank, and background indicates network classification. (C) Mean centralized ranks compared across Literature, Genetic, Genetic 2023+, and Experimental gene sets. Genetic 2023+ gene sets were defined from GWAS published July 27, 2023 or later. Experimental gene sets were defined from publications dated after January 1, 2024. Spearman correlations reported. (D) Box plot illustrating the gene set recovery performance of interaction-type-specific subnetworks from HumanNet and GeneMANIA for the subset of Literature and Genetic gene sets assessed for all type-specific networks. 'ALL TYPES' refers to the full HumanNet and GeneMANIA networks. The center of each box plot represents the median performance, the box boundaries correspond to the upper and lower quartiles, and the whiskers extend to the 5th and 95th percentiles. See Dataset EV2 for full results.

similar (Jaccard = 0.58), indicating a high degree of consistency in physical interaction definitions, especially compared to other interaction types (Fig. EV3E).

Analysis of gene set recovery performance revealed that domain similarity, pathway, and physical interactions were particularly informative across both HumanNet and GeneMANIA (Fig. 3D). In contrast, results for genetic, orthologous, and co-expression interactions were inconsistent between the two sources, reflecting the influence of differing network construction procedures. For example, the higher-performing GeneMA-NIA-orthology network includes interactions predicted from non-human species (Mostafavi et al, 2008), a data source excluded in the latest version of HumanNet (Kim et al, 2022). The HumanNet co-citation network outperformed the full HumanNet interactome (all types) in recovering Literature gene sets, likely highlighting some circularity between the literature-focused network and literature-derived gene sets.

## Consensus networks enhance gene set recovery performance

As part of our earlier systematic evaluation of molecular networks (Huang et al, 2018), we demonstrated that creating consensus networks, such as the Parsimonious Consensus Network (PCNet1.0), could enhance gene set recovery performance. Such an approach leverages the complementary information contained in different databases while excluding non-reproduced interactions. To assess the ongoing benefits of combining interactomes, we evaluated two approaches to assembling consensus networks: "global composites" and "ranked composites" (Fig. 4A). Global composite networks were constructed following the methodology of PCNet1.0, in which a progressive series of composites was constructed by requiring interaction coverage from an increasing number of the 45 interactomes evaluated. Alternatively, we constructed ranked composite networks by varying the number of top interactomes ranked based on median size-adjusted performance. For each ranked composite network threshold, we created two networks requiring interaction coverage from at least two or three interactomes, respectively.

Our evaluation showed that the gene set recovery performance of global composite networks steadily decreased as the interactome coverage threshold became stricter (Fig. 4B). For ranked composites, performance increased as more interactomes were considered, up until 10–15 networks, after which it began to decay. The best overall gene set recovery performance was observed for the ranked composite networks requiring interaction support from two of the top-ranked interactomes. We also evaluated "co-citation-free" ranked composite networks to mitigate the possible confounding effects of co-citation (CC) interactions (Fig. 4C). While less powerful than their CC-inclusive counterparts, these CC-free

consensus networks still generated strong gene set recovery performance results.

From these results, we defined an updated set of Parsimonious Composite Networks (PCNets), balancing performance and parsimony. First, we defined the best-performing ranked composite (top 15 interactomes, 3.85 M interactions) as PCNet2.0. While larger than the latest PCNet1.4 (2.69 M interactions), PCNet2.0 is smaller than many component interactomes, such as STRING (Szklarczyk et al, 2023) and FunCoup (Persson et al, 2021) (Fig. 4D). For situations with computational constraints, we also defined the smaller PCNet2.1 (top 8 interactomes, 1.75 M interactions), and for a CC-free alternative, we defined PCNet2.2 (top 10 CC-free interactomes, 3.32 M interactions). These consensus networks are publicly available via NDEx (Pillich et al, 2021) (ndexbio.org) and include details of supporting interactomes for all interactions.

## Evaluation of predicted interactions and complexes

We leveraged advances in interaction prediction algorithms (Wang et al, 2023) as yet another methodology to evaluate the performance of the panel of interactomes. Using the L3 (paths of length 3) (Kovács et al, 2019) and MPS (Maximum similarity Preferential attachment Score) (preprint: Martini et al, 2021; Wang et al, 2023) algorithms, we predicted interactions using each interactome. The predicted interactions were assessed against held-out interactions from the same network using a 10-fold cross-validation procedure. We further assessed predicted interactions against external sets of physical interactions from multimeric protein complexes recorded in the Comprehensive Resource of Mammalian Protein Complexes (CORUM) database (Giurgiu et al, 2019) and pathway interactions derived from curated, primarily signaling, pathways from the PANTHER knowledgebase (Mi and Thomas, 2009).

To evaluate interactome performance, we focused on the precision of the most confident predictions. Across all interactomes, prediction precision was higher for held-out interactions than external interactions, and MPS generally produced stronger predictions than L3 (Fig. 5A–C). DIP (Xenarios et al, 2000) and Reactome (Gillespie et al, 2022) showed the highest precision for the held-out task, predicting interactions with average precisions at k (P@k) via MPS of 0.57 and 0.55, respectively (Fig. 5A). For CORUM interactions, networks such as Havugimana (Havugimana et al, 2012) (P@k = 0.21, L3) and PTMCode2 (Minguez et al, 2015) (P@k = 0.16, MPS) led in precision (Fig. 5B). In contrast, curated interactomes such as SIGNOR (Lo Surdo et al, 2022) (P@k = 0.05, L3) and Reactome (P@k = 0.04, L3) showed the best performance for predicting PANTHER pathway interactions (Fig. 5C). We further observed that smaller interactomes tended

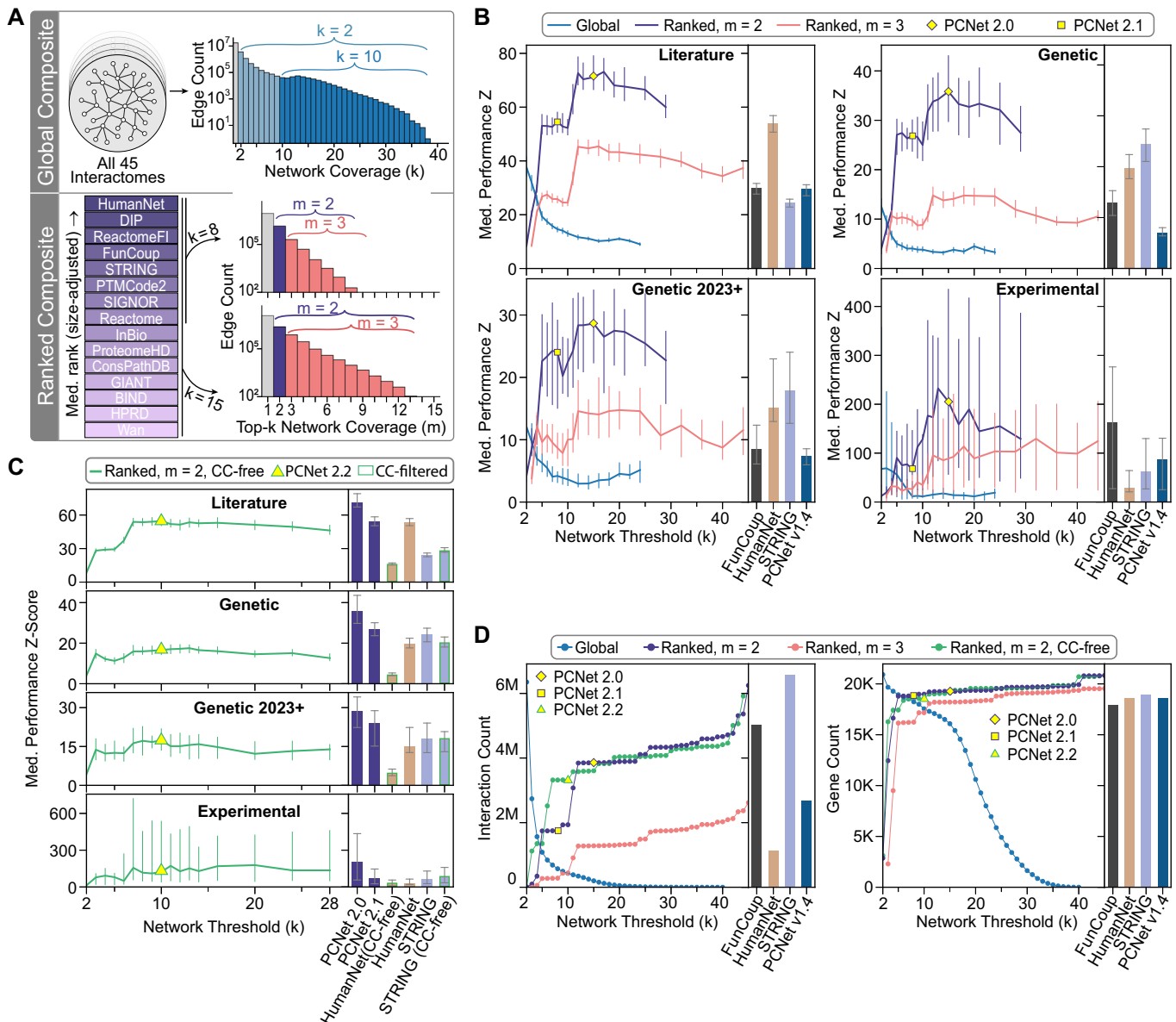

**Figure 4. Definition and evaluation of Parsimonious Composite Networks (PCNets).**

(A) Schematic representation of approaches and thresholds for defining global and ranked composite networks. Global composites include all interactions present in at least $k$ of 45 interactomes. Ranked composites include all interactions present in at least $m$ of the top-$k$ interactomes. (B, C) Median gene set recovery performance for global and ranked composite networks across a range of network thresholds with (B) and without (C) co-citation evidence ("CC-free"). Bar charts show equivalent results for comparison interactomes. In (B), results are compared to top-performing individual networks and PCNet1.4. In (C), results are compared to full and CC-free versions of HumanNet and STRING, as well as PCNet2.0 and PCNet2.1 from the present analysis. All error bars show 95% confidence intervals on the median. The specific number of gene sets evaluated with each network ranges from 820 to 904 for Literature, 75 to 437 for Genetic, 54 to 108 for Genetic 2023+, and 9 to 17 for Experimental. See Dataset EV2 for details and full results. (D) Number of distinct interactions and genes in global and ranked composite networks across a range of network thresholds, compared to top-performing individual interactomes and PCNet1.4.

to predict high-quality interactions, as evidenced by broad support from other networks (Fig. 5D). While all interactomes shared some interactions with the CORUM and PANTHER sets (Fig. EV4A,B), the extent of this overlap did not drive predictive performance for these external interaction sets (Fig. EV4C,D).

In addition to predicting binary interactions, analysis of interactome structures can also be used to predict functional assemblies of genes and proteins. We constructed hierarchical

representations of each interactome by applying the Hierarchical community Decoding Framework (HiDeF (Zheng et al, 2021)). Networks such as ConsensusPathDB (Kamburov et al, 2008), FunCoup (Persson et al, 2021), PROPER (Johnson et al, 2021), TFLink (Liska et al, 2022), and hu.MAP (Drew et al, 2021) identified a large number of small assemblies, while others, such as STRING (Szklarczyk et al, 2023), HumanNet (Kim et al, 2022), and Pathway Commons (Rodchenkov et al, 2020), tended to identify

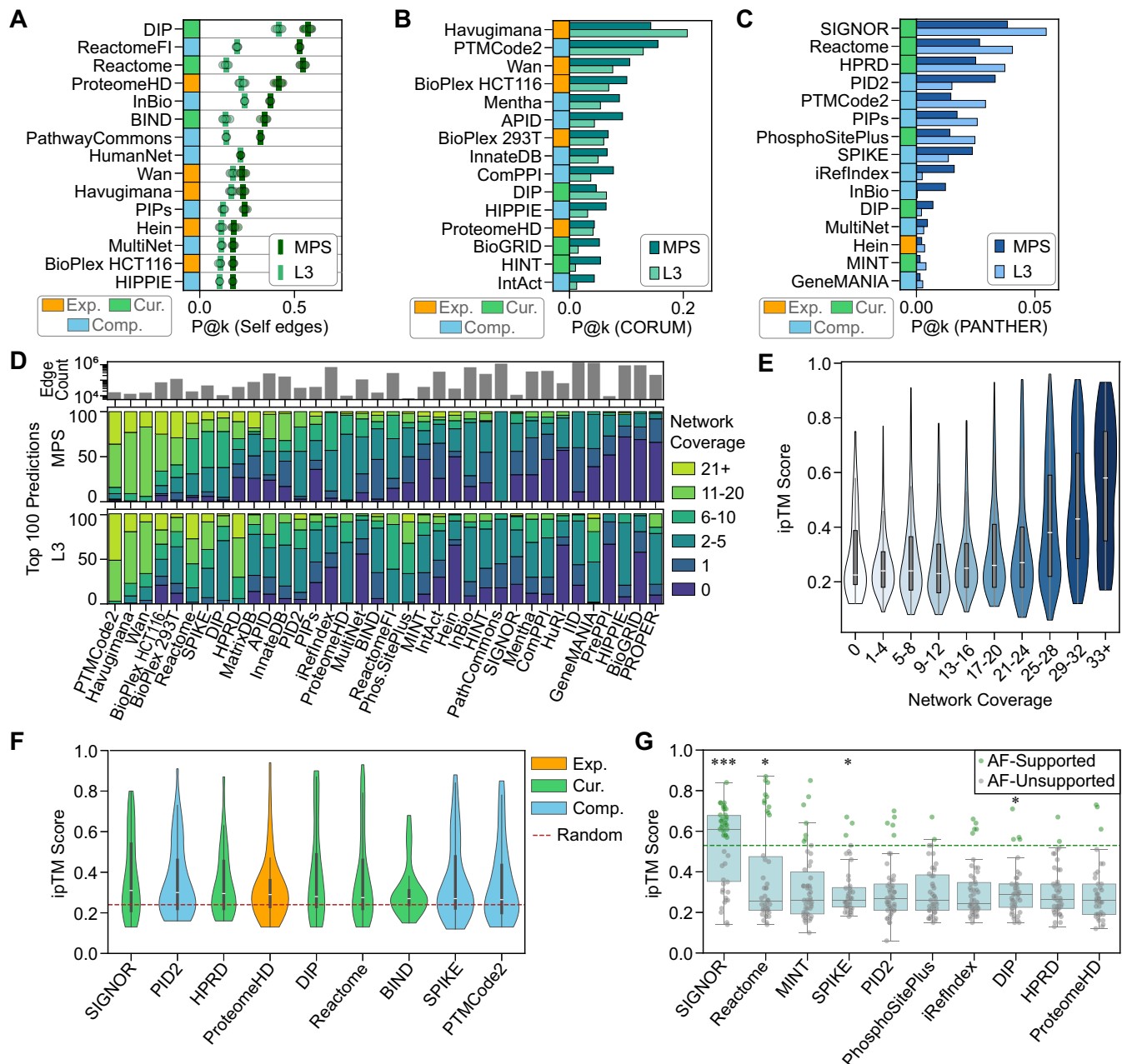

larger assemblies (Fig. EV4E). Interactomes such as DIP (Xenarios et al, 2000) and Havugimana (Havugimana et al, 2012) captured the greatest number of CORUM complexes (Fig. EV4F), while PrePPI (Petrey et al, 2023) and SIGNOR (Lo Surdo et al, 2022) identified complexes with the highest GO annotation similarities among complex members (Fig. EV4G). It should be noted that some interactomes use CORUM or GO during construction, leading to potential biases. Therefore, as an independent metric, we also examined the mean clustering coefficient of predicted complexes, observing the highest clustering for GeneMANIA (Mostafavi et al, 2008), STRING (Szklarczyk et al, 2023), and HumanNet (Kim et al, 2022) (Fig. EV4H). Reactome (Gillespie et al, 2022) and ReactomeFI (preprint: Brunson et al, 2023) were consistently among the top-performing interactomes across all three metrics.

## In silico assessment of predicted interactions

Finally, we utilized AlphaFold-based modeling to assess the interactomes and previously unreported interactions predicted by the MPS algorithm. Though not a substitute for experimental validation, AlphaFold-Multimer (preprint: Evans et al, 2021) provides an in silico approach for assessing potential physical protein interactions. This model builds on the success of AlphaFold (Jumper et al, 2021), a deep-learning model for predicting the structure of individual proteins. AlphaFold-Multimer specifically trains an AlphaFold model using multimeric structures from PDB (wwPDB consortium, 2019) to enable predictions of protein interfaces. Applying AlphaFold-Multimer (AF) to random selections of interactions with varying network coverage, we observed

**Figure 5. Interaction prediction performance and AlphaFold evaluation.**

(A) Interaction prediction evaluation by L3 and MPS algorithms for held-out self-interactions via 10-fold cross-validation. Mean (bar) and individual fold (points) precision at k (P@k, k = size of test set) for the 15 top-performing interactomes based on mean P@k. (B, C) Interaction prediction evaluation by L3 and MPS for external interactions. Precision at k (P@k, k = size of test set) for external interaction sets defined from (B) CORUM complex interactions and (C) PANTHER pathway interactions for the 15 top-performing interactomes with each external interaction set based on mean P@k. (D) Distribution of network coverage for the top 100 interactions predicted by each interactome by MPS and L3. A network coverage of 0 indicates that an interaction was not reported in any of the 45 interactomes. The top bar plot shows the number of edges in the corresponding interactome. Only networks with predictions from both algorithms are shown. (E) AlphaFold-Multimer (AF) interface predicted TM-score (ipTM) for interactions with varying network coverage. The center bar represents the median, the box represents the interquartile range (Q1–Q3), and the upper and lower whiskers represent $Q1 - 1.5IQR$ and $Q3 + 1.5IQR$. The violins extend to the minimum and maximum observations, and represent 50 randomly sampled interactions per network coverage value. (F) AF ipTM scores for samples of 50 interactions per network. Interactomes shown are enriched for interactions with high interface scores ($q < 0.15$, Mann–Whitney U-test, BH correction), as compared to a background distribution of 1779 randomly generated protein pairs. The center bar represents the median, the box represents the interquartile range (Q1–Q3), and the upper and lower whiskers represent $Q1 - 1.5IQR$ and $Q3 + 1.5IQR$. The violins extend to the minimum and maximum observations, and represent 50 interactions per network. See Appendix Fig. S3B for results for all interactomes. (G) AlphaFold-Multimer assessment of 50 previously unreported interactions per network predicted using MPS. The top 10 interactomes ranked by mean ipTM score of previously unreported interactions are shown. AF-supported interactions were defined as protein pairs with an interface score in the 95th percentile (ipTM > 0.53, dashed line) of the background distribution of 1779 randomly generated protein pairs. The distribution of ipTM scores for previously unreported interactions was assessed against the distribution of scores from randomly generated protein pairs by a Mann–Whitney U-test (*$q < 0.1$, ***$q < 10^{-5}$, BH correction). The center of each box plot represents the median score, the box boundaries correspond to the lower (Q3) and upper (Q3) quartiles, and the whiskers extend to $Q1 - 1.5IQR$ and $Q3 + 1.5IQR$. See Appendix Fig. S3D for results from all interactomes. See Dataset EV3 for full interaction prediction results and Dataset EV4 for full AF results.

higher interface predicted template modeling (ipTM) scores for interactions present in more than 25 interactomes (Fig. 5E), especially for the most supported interactions (≥33 supporting interactomes). This pattern was consistent with higher proportions of experimentally resolved structures amongst highly supported interactions (Appendix Fig. S3A). We further identified nine interactomes containing interactions with significantly higher ipTM scores than randomly generated protein pairs (Fig. 5F, $q < 0.15$). Again, we observed a correlation between the presence of experimentally resolved structures in PDB and ipTM scores, though these generally accounted for fewer than 20% of tested interactions (Appendix Fig. S3B,C).

Next, we applied AF to assess the top 50 previously unreported interactions predicted by each interactome via MPS. We identified 126 distinct interactions with an ipTM score in the 95th percentile of scores from randomly generated protein pairs (ipTM > 0.53), which we defined as AF-supported interactions. The ipTM scores of predicted interactions were significantly higher than randomly generated protein pairs for SIGNOR (Lo Surdo et al, 2022), Reactome (Gillespie et al, 2022), SPIKE (Paz et al, 2011), and DIP (Xenarios et al, 2000) (Fig. 5G, Appendix Fig. S3D). While the AF status of an interaction was not independent of the existence of experimentally resolved structures ($p = 0.048$, $\chi^2$ test), we observed that the AF-supported interactions were, in fact, less likely to involve proteins with existing structures. Specifically, 124/126 AF-supported interactions involved at least one protein with no structure in PDB (Appendix Fig. S3E). The AF-supported interactions represented functionally coherent pairs (Fig. EV5), and the proteins involved showed enrichments for G-protein coupled receptor ($q = 9.9 \times 10^{-12}$), GTPase ($q = 1.1 \times 10^{-6}$) and hydrolase ($q = 7.5 \times 10^{-4}$) activities. Interactions predicted by SIGNOR (Lo Surdo et al, 2022) showed particularly high ipTM scores, with 64% being AF-supported, indicating that many of these predictions likely represent true physical interactions.

## Discussion

As the range and scope of molecular networks and network databases increase, it is essential to continually evaluate and assess these valuable resources. Here, we have presented the most expansive snapshot of

network resources to date (Fig. 1), assessing the current state of a diverse range of 45 interactomes, including their features, contents, and performance for discovering disease genes and prioritizing novel interactions. Our analysis highlights the leading interactomes across various applications (Table 1), revealing significant differences depending on the task at hand. Many of these interactomes now cover nearly all protein-coding genes (Fig. 2A), though not all biological domains are equally represented, with biases particularly prevalent amongst experimental networks (Fig. 2C,D, Table 1). Consequently, biological representation should be carefully considered when selecting a network. While some underrepresentation may reflect genes with truly fewer interactions, these results suggest a need for increased focus on less-studied proteins and functions such as transporter and receptor activity.

As interactomes have become more numerous, understanding the utility of different data sources and network structures has become critical for network biology applications, such as disease gene prioritization (Huang et al, 2018; Kim et al, 2022; Mosca et al, 2021). A small set of networks, including HumanNet (Kim et al, 2022), STRING (Szklarczyk et al, 2023), and FunCoup (Persson et al, 2021), consistently produced the strongest disease gene set recovery performance across Literature and GWAS-derived Genetic gene sets (Fig. 3B, Table 1). However, differences in performance across gene set sources highlighted the importance of careful network selection. Performance was consistently higher for Literature gene sets than Genetic gene sets, possibly due to a combination of gene set size and the shared reliance on publications between the Literature gene sets and the sources used to construct many of the interactomes. However, the interactome rankings remained stable when considering recent genetic findings and experimentally derived gene sets (Fig. 3C).

Consistent with previous observations (Huang et al, 2018; Mosca et al, 2021), gene set recovery performance was highly correlated with the number of interactions in each network. After adjusting for this size effect, we found that HumanNet (Kim et al, 2022) and DIP (Xenarios et al, 2000) had the highest performance per interaction, indicating that the information in these networks is of very high quality (Fig. 3B). In contrast, some high-performing interactomes' rankings suffered after size adjustment, indicating that they may contain many lower-quality interactions that are

**Table 1.** Summary of interactome recommendations across network biology applications.

| Outcome/Analysis | | Top Network Choices | Important Considerations |
|---|---|---|---|
| Best Gene Coverage | Experimental | BioPlex293T, BioPlex HCT116, HuRI | • Many databases continue to expand |
| | Curated | BioGRID, HINT, HPRD | • Experimental and curated interactomes contain high-quality interactions but are more incomplete |
| | Composite | HIPPIE, GeneMANIA, Pathway Commons | • Composite interactomes depend on a wide variety of sources |
| Biological Representation | Least Citation and Expression Bias | GeneMANIA, STRING, HumanNet | • Bias towards highly cited and highly expressed genes reflects a combination of experimental and curation biases, and gene functional importance |
| | Highest Citation Bias | PID2, DIP, PhosphoSitePlus | • Networks should be examined for coverage of relevant tissues and biological functions |
| | Highest Expression and Abundance Bias | Havugimana, ProteomeHD, Wan | |
| | Gene Function Prediction | HumanNet, Havugimana, Reactome (GO:BP), PTMCode2 (GO:CC), ReactomeFI (GO:MF) | |
| Best Disease Gene Prioritization | Literature | HumanNet, ConsensusPathDB, PCNet2.0 | • Large networks demonstrate higher performance using network propagation due to better coverage of genes of interest and a greater range of prior knowledge |
| | Genetic | STRING, HumanNet, FunCoup, PCNet2.0 | • Certain interaction types, such as co-citation and domain similarity, contribute strongly to performance |
| | Excluding co-citation interactions | FunCoup, PCNet2.2, STRING (CC-free) | |
| Best Interaction Prediction | Self-interactions | DIP, ReactomeFI, Reactome, ProteomeHD | • Smaller networks with high-quality and relevant interaction types perform better |
| | CORUM interactions | Havugimana, PTMCode2, ReactomeFI | • Interaction prediction with large networks can become computationally intractable |
| | PANTHER interactions | SIGNOR, Reactome, HPRD | • Some interactomes incorporate CORUM and PANTHER |
| Best Protein Assembly Prediction | CORUM Recapitulation | DIP, Havugimana, Reactome | • Different network topologies lead to different hierarchical assembly structures |
| | Functional Coherence | PrePPI, SIGNOR, TFLink | • Some interactomes incorporate CORUM and GO during network construction |
| Best in silico assessment (AlphaFold) | Self-interactions | SIGNOR, PID2, HPRD, ProteomeHD | • Combining interaction prediction with in silico assessment may help address systematic gaps in interactomes |
| | Previously unreported interactions | SIGNOR, SPIKE, Reactome | • Experimental validation remains necessary for confirming predicted interactions |

nevertheless overcome by the network propagation procedure. We found that some low-quality and potentially spurious interactions can be excluded by selecting only those interactions present in multiple sources. In this vein, we used the size-adjusted interactome rankings to construct high-quality parsimonious composite networks (PCNets), requiring all interactions to be supported by at least two databases. Despite our approach not explicitly accounting for redundancy between sources (Melkonian et al, 2022), the higher gene set recovery performance (Fig. 4) indicates an improvement in true biological signal. These networks, PCNet2.0, PCNet2.1, and PCNet2.2, are publicly available via NDEx.

Further, we developed an additional evaluation metric to assess the quality of available interactomes based on their capacity for interaction prediction. In contrast to gene set recovery results, the precision of interaction prediction was not driven by interactome size, and overall interactome rankings for interaction prediction varied substantially from rankings for gene set recovery (Fig. 5A–C, Table 1). In particular, the top-performing networks included many with improved size-adjusted performance in gene set recovery, indicating the quality rather than quantity of interactions is crucial for interaction prediction. Our in silico assessment with AlphaFold-Multimer further reinforced the importance of network selection, with a small subset of interactomes such as SIGNOR (Lo Surdo et al, 2022) and Reactome (Gillespie et al, 2022) predicting interactions with high AlphaFold-Multimer support (Fig. 5F, Table 1). While experimental physical protein–protein interaction networks such as Havugimana

(Havugimana et al, 2012) remain optimal for predicting stable multiplex complexes (Figs. 5B and EV4F), our results suggest that networks incorporating signaling and pathway interactions could be more frequently utilized to predict binary protein interactions (Fig. 5G). The majority of previously unreported interactions supported by our AlphaFold-Multimer analysis involved receptor and regulatory interactions, suggesting that in silico approaches may indeed help fill gaps in interactomes caused by experimental limitations related to transmembrane proteins and dynamic interactions (Fig. 2D). On the other hand, the absence of in silico support for other predicted interactions should not be considered evidence against their existence. AlphaFold-Multimer specifically models direct physical interactions and thus does not capture the full range of potential gene-gene relationships. Additional in silico and in vitro validation of these predicted interactions is essential to build on the evaluations presented here.

We must also acknowledge the contributions of other studies to understanding the current state of interactomes. Here, we have used a biological lens to understand the information contained within human molecular networks and their performance across a broad range of applications. This work, therefore, complements other detailed studies that have focused on network topology (Mosca et al, 2021; Ramos et al, 2024), re-wiring (Huttlin et al, 2021), and disease and tissue context (Greene et al, 2015; Zitnik and Leskovec, 2017; Li et al, 2024) across interactomes. Additionally, interactome data are increasingly used to provide machine learning

features and determine the architecture of neural networks. These applications warrant a specific and detailed analysis of interactome resources beyond our study.

We anticipate that molecular network data generation and analysis will continue apace, making regular evaluations of available networks essential for optimizing biological discovery. This work provides both an up-to-date benchmarking of available interactomes and a set of refined and expanded tools for ongoing evaluation of biological networks in the future (https://github.com/sarah-n-wright/Network_Evaluation_Tools). While no framework can evaluate all possible interactomes for all applications, we hope this work will serve as a broad guide for network selection, and we welcome the continued development of complementary approaches.

# Methods

### Reagents and tools table

| Reagent/ Resource | Reference or Source | Identifier or Catalog Number |
|---|---|---|
| **Experimental Models** | | |
| N/A | | |
| **Recombinant DNA** | | |
| N/A | | |
| **Antibodies** | | |
| N/A | | |
| **Oligonucleotides and other sequence-based reagents** | | |
| N/A | | |
| **Chemicals, Enzymes and other reagents** | | |
| N/A | | |
| **Software** | | |
| neteval 0.2.2 | https://github.com/sarah-n-wright/Network_Evaluation_Tools, this study | |
| BEDOPS 2.4.41 | https://bedops.readthedocs.io/ (Neph et al, 2012) | |
| NDEx2 Python Client 3.5.0 | https://github.com/ndexbio/ndex2-client (Pillich et al, 2021) | |
| EGAD 1.32.0 | https://doi.org/doi:10.18129/B9.bioc.EGAD (Ballouz et al, 2017) | |
| L3 1.0.2 | https://github.com/kpisti/L3 (Kovács et al, 2019) | |
| MPS | https://github.com/spxuw/PPI-Prediction-Project/ (Wang et al, 2023) | |
| HiDeF 1.1.5 | https://github.com/fanzheng10/HiDeF (Zheng et al, 2021) | |
| ColabFold 1.5.5 | https://github.com/YoshitakaMo/localcolabfold (Mirdita et al, 2022) | |
| Python 3.10 | https://www.python.org/ | |
| R 4.4 | https://www.r-project.org | |
| Adobe Illustrator CC 2024 | https://www.adobe.com/ | |
| Scipy 1.7.2 | https://scipy.org/ (Virtanen et al, 2020) | |

| Reagent/ Resource | Reference or Source | Identifier or Catalog Number |
|---|---|---|
| Networkx 2.6.3 | https://networkx.org/ | |
| Statsmodels 0.13.5 | https://www.statsmodels.org/ (Seabold and Perktold, 2010) | |
| goatools | https://github.com/tanghaibao/goatools (Klopfenstein et al, 2018) | |
| Cytoscape v3.10.2 | https://cytoscape.org/ (Shannon et al, 2003) | |
| **Other** | | |
| N/A | | |

## Methods and protocols

### Data collection and processing

**Collection and standardization of interaction data**. Where possible, all networks were downloaded from primary sources (Appendix Table S1). DIP (Salwinski et al, 2004) and BIND (Bader et al, 2003) were downloaded from PathwayCommons v12 (Rodchenkov et al, 2020), and PCNet 1.4 (Huang et al, 2018) and PID v2.0 (Pillich et al, 2023) were downloaded from NDEx (Pratt et al, 2017; Pillich et al, 2021) (ndexbio.org). All non-human interactions were excluded, except in cases where the authors used orthologous interactions to enhance human networks (Fig. 1A). All networks were standardized to:

- Remove duplicate interactions.
- Remove self-interactions.
- Binarize any interactions involving more than two genes by defining edges between all pairs of genes.
- Convert all interactions to undirected.

The GeneMANIA (Mostafavi et al, 2008), GIANT (Greene et al, 2015), and hu.MAP 2.0 (Drew et al, 2021) interactomes contained more than 15 M interactions. Therefore, we filtered these networks to the top 10% of interactions using the provided interaction scores. For HumanNet, we used the functional network extended by co-citation (HumanNet-XC) as the primary interactome and defined the functional subnetwork (HumanNet-FN) as the co-citation-free HumanNet (CC-free). The STRING co-citation-free (CC-free) network was defined by excluding interactions supported solely by the "textmining" channel.

**Collation of network metadata**. Network dependencies and interaction types were collated from publicly available information based on definitions in Appendix Table S2. Dependencies and interaction types were manually curated from associated publications and websites, as well as metadata available within the interactome data files (such as "Source," "Database," and "Interaction Type" columns). Where available, PMIDs associated with interactions were used to identify experimental sources. A network dependency between a target and a source network was defined as a relationship where interactions from the source network were directly incorporated into the target network. Using a network to train, prioritize, or score interactions was not considered a dependency.

**Standardization of gene identifiers**. We mapped all gene identifiers to NCBI Gene IDs using APIs from MyGeneInfo (Wu et al, 2013; Xin et al, 2016), UniProt (UniProt Consortium, 2023), HGNC (Seal et al, 2023), and Ensembl (Cunningham et al, 2022). Input identifiers were first updated using the most relevant database to address out-of-date identifiers and then converted to NCBI Gene IDs. For mapping HGNC Gene Symbols, previously approved symbols were prioritized over alias symbols. The performance of our gene mapping pipeline was compared to the use of MyGeneInfo alone, achieving a reduction of over 60% in genes that could not be mapped to NCBI gene identifiers (Appendix Fig. S4A).

**Collation and processing of gene metadata**. We sourced gene and protein annotation features from HGNC (Seal et al, 2023) (chromosome, locus type, locus group) and NCBI (Sayers et al, 2022) (citation count) on December 20, 2023. The set of protein-coding genes was defined by the HGNC Locus Group "protein-coding gene." Functional Gene Ontology (GO) associations were downloaded from NCBI using goatools (Klopfenstein et al, 2018) on March 29, 2023.

To assess mRNA expression patterns, we sourced and processed median gene-level TPM by tissue from GTEx v8 (Data ref: GTEx Portal, 2017):

- For genes with multiple transcripts, we consolidated the values for all transcripts within each tissue. We took the mean of all transcripts if all values had a similar magnitude (all TPM observations above or below $10^{-4}$); otherwise, we took the maximum.
- Overall expression levels were calculated as the mean across all tissues.

For protein abundance, we utilized normal tissue abundance values by tissue reported by the Human Protein Atlas (HPA) v23 (Uhlén et al, 2015, Data ref: The Human Protein Atlas, 2023).

- Entries with "Uncertain" reliability were excluded.
- Tissues with fewer than 1000 observations were excluded.
- The categorical abundance levels provided by HPA were converted to numerical values (Not detected = 0, Low = 1, Medium = 2, High = 3).
- Converted numerical values for each gene were consolidated by taking the mean of associated protein abundance values across all tissues.

Positional gene conservation scores (phyloP) were sourced from the UCSC Genome Browser (Nassar et al, 2023, Data ref: UCSC Genome Browser, 2017), calculated using the PHAST package for multiple alignment of 29 vertebrate species to the hg38 human genome.

- PhyloP scores from the reference chromosomes were aggregated for each transcript using *BEDOPS v2.4.41* (Neph et al, 2012) based on coding sequences (CDS) derived from GENCODE (Frankish et al, 2019) v46 Basic Gene Annotation (Data ref: GENCODE, 2024).
- The final gene conservation score was defined per gene as the mean phyloP across all positions within the gene's CDS.

Lists of experimentally resolved structures in the Protein Data Bank (PDB) (wwPDB consortium, 2019) were downloaded on August 14, 2024.

- Interactions with experimentally resolved structures were broadly defined as any protein pair reported as part of a human complex with at least two distinct protein entities that could be mapped to NCBI Gene IDs via UniProt Accession Codes.
- Proteins with individual experimentally resolved structures were broadly defined as any protein identified from structures containing one distinct protein entity (including homodimers and partial structures) that could be mapped to NCBI Gene IDs via UniProt Accession Codes.

**Definition of gene and interaction sets**. We generated collections of gene sets from three sources to evaluate gene set recovery performance. All gene identifiers were converted to NCBI Gene IDs.

- Literature: we obtained literature-curated disease gene sets from DisGeNET (Piñero et al, 2017) via the disgenet.com API, utilizing BEFREE (Bravo et al, 2015) gene-disease associations sourced from the text-mining of MEDLINE abstracts. All data was downloaded on December 22, 2023. We retained sets with a maximum of 500 genes and at least 5 genes in every interactome to give 906 Literature sets.
- Genetic: we created genetic gene sets from genome-wide association studies (GWAS) via the GWAS Catalog (Sollis et al, 2023, Data ref: GWAS Catalog, 2024). All data was downloaded on January 22, 2024. From the full download, we extracted all SNPs mapped to an NCBI Gene ID with a significant association ($p$-value $< 5 \times 10^{-8}$) to a phenotype in the Experimental Factor Ontology (EFO). We created gene sets for all phenotypes with fewer than 500 distinct gene associations across all studies to give 699 Genetic gene sets.
- Genetic 2023+: we created a collection of recent genetic gene sets using only GWAS with a publication date after July 27, 2023, thus postdating the latest update of interactomes incorporating co-citation interactions. This produced 48 sets which we termed the "Genetic 2023+" gene sets.
- Experimental: we identified gene sets generated from experimental data published between January 1 and January 24, 2024. Candidate studies were identified from PubMed using grouping keywords such as "module," "profile," "gene set," and "component," combined with experimental context keywords such as "transcriptional," "regulatory," "differentially expressed," "biomarker," "single-cell," and "gene expression." Studies were selected if they generated publicly available gene sets between 20 and 250 genes (or data from which these could be readily defined) and did not utilize literature or network resources for set definition, resulting in 17 Experimental gene sets (Appendix Table S3).

To analyze each interactome, we selected gene sets with a minimum of 20 (Literature, Genetic) or 10 (Genetic 2023+, Experimental) genes present in the interactome. All gene sets are available in Dataset EV5.

External sets of complex and pathway interactions were sourced from CORUM (Giurgiu et al, 2019) and PANTHER (Mi and Thomas, 2009) via PathwayCommons (Rodchenkov et al, 2020). Interactions were standardized using the same procedure as for interactomes to generate interaction lists.

**Creation of interaction-type-specific networks**. We defined interaction-type-specific subnetworks from HumanNet (Kim et al, 2022) and GeneMANIA (Mostafavi et al, 2008). From HumanNet, we downloaded HumanNet-co-citation (HS-CC), HumanNet-coexpression (HS-CX), HumanNet-pathway (HS-DB), HumanNet-domain (HS-DP), HumanNet-genetic interaction (HS-GI), HumanNet-phylogenetic similarity (HS-PG), and HumanNet-physical (HS-PI). These type-specific HumanNet networks were constructed via combinations of curated and experimental sources under a supervised learning framework. All HumanNet networks were downloaded from https://staging2.inetbio.org/humannetv3/ and standardized per our pipeline outlined above. From GeneMANIA, we downloaded source files based on assigned types ("Co-expression," "Genetic_Interactions," "Pathway," "Physical_Interactions," "Predicted," and "Shared_protein_domains") from https://genemania.org/data/current/Homo_sapiens/. All interactions within a source were concatenated to form interaction-type-specific GeneMANIA networks. GeneMANIA sources for genetic and co-expression interactions include large-scale screens with interaction scores. Therefore, using the maximum interaction score for each distinct interaction, we filtered these interaction-type-specific networks to the top 10% of genetic interactions and the top 3% of co-expression interactions. While drawing on different data types, we assigned HumanNet-phylogenetic similarity and GeneMANIA-predicted to the category "Orthology" as both utilized information from non-human species. Interaction-type-specific network similarities were calculated as the Jaccard similarity of undirected interactions after network processing.

**Deposition of interactomes to NDEx**. All standardized source networks were uploaded to the NDEx network set "State of the Interactomes: source networks" via the NDEx2 Python Client (Pillich et al, 2021) v3.5.0. All genes were indexed by NCBI Gene IDs and annotated with current approved HGNC Symbols as of April 2, 2024. The original gene identifiers used by the source database were maintained, as were additional edge annotations where available, such as PubMed IDs, interaction type, or detection method. PCNet2.0, PCNet2.1, and PCNet2.2 were similarly uploaded via the NDEx2 Python Client. PCNet genes were annotated with the current approved HGNC Symbols, and interactions were annotated with the number of supporting interactomes and a list of those supporting interactomes. All network figures were generated using Cytoscape (Shannon et al, 2003).

NDEx is an open-source, publicly available software infrastructure that facilitates the storage, exchange, visualization, and publication of network models and data among scientists. Thanks to its full integration with the Cytoscape desktop application, users can employ NDEx to import, export, and analyze biological networks using the large variety of tools and applications available in the Cytoscape ecosystem. The platform's key advantages include fostering collaborative research through shared networks, enabling the reproducibility of scientific findings, and promoting the discovery of new biological insights by integrating disparate data types into comprehensive network models. NDEx thus offers a centralized resource to enhance the utility and accessibility of network-based data and analyses in elucidating complex biological systems. To learn more about NDEx, please review the FAQ page at www.ndexbio.org.

### Representation analysis

**Gene-level annotations**. For each quantitative gene-level annotation (citation count, mRNA expression, protein abundance, and gene conservation), we assessed whether each interactome contained genes with greater median annotation values different than would be expected by chance using a permutation test.

1. Calculate the median annotation value for each interactome as the median value for protein-coding genes contained in that interactome. The set of protein-coding genes was defined by the HGNC Locus Group "protein-coding gene."
2. For each of 10,000 random samples, take a sample without replacement of size N from the set of all protein-coding genes, where N is the number of genes in the interactome. Take the median annotation value of all genes in the sample.
3. Generate empirical one-sided $p$-values by calculating the number of permuted medians that are greater than the observed median. If no permuted medians are greater than the observed median, assign a $p$-value of $1 \times 10^{-4}$.
4. Calculate Bonferroni corrected $q$-values based on the number of interactomes $q = min(45 \cdot p, 1)$.

Global correlations between network coverage and gene features were calculated using Spearman correlation via scipy (Virtanen et al, 2020). Where the reported $p$-value is lower than the precision of the test we report $p = 1 \times 10^{-50}$. For mRNA expression all genes with mRNA data were binned into 20 percentile bins based on the mean expression across all tissues. Due to the number of genes with near-zero average mRNA expression, the lowest 20th percentile was treated as a single bin for mRNA expression. For protein abundance, all genes with protein abundance data were binned into 20 percentile bins based on the mean abundance across all tissues. All correlations were calculated for protein-coding genes only. Citation counts were adjusted for mean mRNA expression using a log–log ordinary least squares linear regression via statsmodels (Seabold and Perktold, 2010), excluding genes with zero citations or a mean mRNA expression of zero.

Sets of tissue-enhanced genes for mRNA expression and protein abundance were defined using the criteria outlined by the Human Protein Atlas (HPA) (Uhlén et al, 2015, Data ref: The Human Protein Atlas, 2023). Using mRNA Expression data from GTEx (Data ref: GTEx Portal, 2017), all protein-coding genes were classified as one of:

(a) Low Expression—TPM < 1 in all tissues.
(b) Tissue Enriched—five-fold higher TPM in one tissue compared to all other tissues.
(c) Group Enriched—five-fold higher TPM in a group of 2–7 tissues compared to all other tissues.
(d) Tissue Enhanced—five-fold higher TPM in one tissue compared to the average of all other tissues.
(e) Broadly Expressed—all genes not otherwise classified.

The genes classified as Tissue Enriched, Group Enriched, or Tissue Enhanced for a given tissue were considered part of the tissue-enhanced mRNA expression gene set for that tissue. All protein-coding genes were similarly classified based on the quantified HPA protein abundance levels (see the "Collation and processing of gene metadata" section) of associated proteins. Low Abundance was defined as abundance <0.5, and Tissue Enriched, Group Enriched, and Tissue Enhanced genes were defined based on three-fold, rather than five-fold, higher abundance levels. All genes classified as Tissue Enriched, Group Enriched, or Tissue Enhanced for a given tissue were considered part of the tissue-enhanced protein abundance gene set for that tissue. Tissues

with more than 10 genes assigned were maintained for analysis. The set of genes in each interactome (filtered to those present in the mRNA expression or protein abundance data, respectively) was tested for enrichment of tissue-enhanced genes using a two-sided Fisher's Exact Test via scipy (Virtanen et al, 2020) with BH correction via statsmodels (Seabold and Perktold, 2010). The number of genes with reported mRNA or protein levels (after conversion to NCBI Gene IDs) was used as the background for the Fisher's Exact tests.

**Interaction-level annotations.** To assess the relationships between gene annotation features and the presence of interactions across all networks, we assessed the density of interactions between genes with varying citation count, mRNA expression, protein abundance, and chromosome number. Densities were calculated between genes binned based on annotation value using the following procedure:

1. Construct annotation gene bins based on the annotation values for all genes that are protein-coding or present in at least one interactome. Previously calculated percentile bins are used for mRNA expression and protein abundance (see the "Gene-level annotations" section). Fifty bins of approximately equal size are constructed for gene citation count, such that all genes with equal citation counts are in the same bin. Chromosome bins are defined by chromosome number.
2. Assign every interaction *GeneA-GeneB* present in at least one interactome to two bins: BinA corresponding to the annotation value for GeneA and BinB corresponding to the annotation for GeneB.
3. For each combination of BinA and BinB, count the number of distinct interactions ($n_{AB}$) between genes in BinA and genes in BinB across all 45 interactomes.
4. Normalize each observed count of interactions based on the number of possible interactions to give the interaction density $D$ (Eq. 1). The number of possible interactions between BinA and BinB is calculated based on genes that are present in at least one interactome, where $A$ represents the set of genes in BinA, and $B$ represents the set of genes in BinB:

$$D_{AB} = n_{AB} \cdot \left( \frac{|A||B|}{2} \right)^{-1} \text{ if } A \neq B, \text{ else } D_{AA} = \frac{n_{AA}}{\binom{A}{2}} \qquad (1)$$

**Gene and interaction functional annotations.** To assess the representation of biological processes, molecular functions, and cellular compartments, we calculated gene set enrichment for GO Slim terms (https://current.geneontology.org/ontology/subsets/goslim_generic.obo, accessed on June 19, 2023) using a two-sided Fisher's Exact Test via scipy (Virtanen et al, 2020) with BH correction via statsmodels (Seabold and Perktold, 2010).

- From the GO Slim ontology, we selected terms with between 100 and 4000 associated genes and excluded any terms that were parents of other terms in the group, resulting in a group of 93 terms.
- The background for the enrichment analysis was set to all protein-coding genes as defined by HGNC with at least one GO association.
- All genes associated with a term were also considered to be associated with all parents of that term.

The interaction density (Eq. 1) was calculated within and between the group of 93 GO Slim terms, where each term represented a bin, and genes were permitted multiple bin associations. The denominator of Eq. 1 was set based on the number of unique pairwise combinations of genes between GO terms, accounting for overlapping genes between GO terms.

**Gene function prediction via guilt-by-association.** Guilt-by-association analysis was performed using Extending Guilt by Association by Degree (Ballouz et al, 2017) (EGAD), as implemented in the R package EGAD v1.32.0. Briefly, this method utilizes a neighbor-voting framework and compares the results to a baseline prediction based solely on the degree of genes within the network. We identified a broad group of GO annotation gene sets from GO associations from NCBI (see the "Collation and processing of gene metadata" section). Again, all genes associated with a term were assumed to be associated with all parent terms. We performed the guilt-by-association analysis using 5-fold cross-validation, holding out one-fifth of genes in each GO set per fold, followed by calculation of the AUPRC for prediction of the held-out genes. Null AUPRC values were reported for each fold using predictions based on genes ranked by node degree, acting as a measure of degree bias (Ballouz et al, 2017). For each interactome, the observed AUPRC values were averaged over all GO terms with more than 10 and fewer than 250 genes present in the interactome.

### Gene set recovery evaluation

**Network propagation with subsampling.** Gene set recovery via network propagation was implemented following the procedure previously established (Huang et al, 2018) for each interactome.

1. Curate gene sets (see the "Definition of gene and interaction sets" section) and take the intersection of each gene set and the interactome genes.
2. Randomly subsample a proportion $\rho$ without replacement from this intersection as seed genes, with the remaining portion of the gene set becoming the held-out set.
3. Create the binary vector $F_0$ representing each gene's initial seed gene status.
4. Propagate from the seed genes through the network using a random walk with restart model (Vanunu et al, 2010), utilizing the closed-form solution (Leiserson et al, 2014):

$$F = (1 - \alpha)F_0 \cdot (I - \alpha A_{norm})^{-1} \qquad (2)$$

Where $F$ is the vector of propagation scores for all genes, $\alpha$ is the network propagation constant, and $A_{norm}$ is the degree-normalized adjacency matrix of the interactome.
5. Define the held-out set $h$ as the true positives, and rank all non-seed interactome genes by propagation score. Then, calculate the area under the precision-recall curve (AUPRC) for recovering $h$.
6. Repeat the subsampling, propagation, and AUPRC calculation steps for 50 gene set samples and average the observed AUPRC values to give a mean AUPRC for each gene set.
7. Construct 50 randomized versions of each interactome via degree-matched edge shuffling and repeat steps 2–6 for each gene set to create a null distribution of mean AUPRC values.
8. Calculate the "Performance" score, defined as the robust Z-statistic (Rousseeuw and Croux, 1993) between the observed and null mean AUPRC values for each gene set. This statistic uses the median absolute deviation (MAD) as a scale estimator, making it more robust to outliers and more reliable for non-normally distributed data.

9. Calculate the "Performance Gain" for each gene set, defined as the difference between the observed mean AUPRC and the median mean AUPRC from the associated null networks, normalized by the median mean AUPRC from the associated null networks.
10. Repeat steps 1–9 for all interactomes.
11. Calculate size-adjusted performance metrics by fitting a linear regression model for each gene set between the $\log_{10}$ interaction count of each interactome and the observed performance scores. The residual for each interactome is the size-adjusted performance of the interactome for that gene set.

**Parameter optimization for gene set recovery.** Previous studies have shown that the optimal gene set recovery parameters can vary between networks (Huang et al, 2018). Therefore, to enable assessment of interactomes under their best conditions, we performed optimization of the two key gene set recovery parameters $\alpha$ (network propagation constant) and $\rho$ (subsampling proportion) for each interactome and gene set using the 50 MSigDB Hallmark pathways (Liberzon et al, 2015) as an independent source of gene sets. The Hallmark pathways provide a source of high-quality gene sets, independent of the larger and disease-focused Literature, Genetic, and Experimental gene sets used for evaluation. We compared gene set recovery performance across intervals of $\alpha \in \{0.2, 0.3,…0.9\}$ and $\rho \in \{0.1, 0.0,…0.8\}$ for all interactomes and MSigDB gene sets (Appendix Fig. S4B, Dataset EV6) and used a linear regression of mean performance using Huber Regression (Huber, 1964) via sklearn (Pedregosa et al, 2011) with $\alpha$, $\rho$, network size ($\log_{10}$ of normalized interaction count) and gene set coverage (normalized size of intersection between gene set and interactome genes) as features. This analysis identified that the subsampling parameter $\rho$ had a stronger influence on performance than the network propagation constant $\alpha$. To identify the optimal parameters, we first calculated the average optimal parameter values for each interactome-gene set pair by taking the mean of the five top-performing parameter sets. We then fit the average optimal subsampling parameter to the normalized network size and gene set coverage (Appendix Fig. S4C). The resulting formula was used to set the subsampling parameter:

$$\rho = 0.44 + 0.0093 \cdot S - 0.0013 \cdot C \quad (3)$$

Where $S$ is the $\log_{10}$ network size (interaction count), and $C$ is the gene set coverage. We restricted the value to $0.1 < \rho < 0.8$. Next, we fit the optimal propagation constant for each interactome to the normalized network size $S$, the average subsampling parameter $\overline{\rho}$, and the average gene set coverage $\overline{C}$ across all MSigDB gene sets (Appendix Fig. S4D). The resulting formula was used to set the propagation constant for interactomes:

$$\alpha = 0.59 + 0.24 \cdot \overline{\rho} - 0.058 \cdot S + 0.00036 \cdot \overline{C} \quad (4)$$

Where $\overline{\rho}$ is the mean subsampling parameter across all gene sets, $S$ is the $\log_{10}$ network size (interaction count), and $\overline{C}$ is the mean gene set coverage across all gene sets. We restricted the value to $0.2 < \alpha < 0.9$. By setting the parameters in this way, we reduce the chance that gene set recovery performance is driven by variation in how close the chosen parameters are to the optimal for each interactome. For type-specific, global composite, and ranked composite networks, we set the subsampling parameter using Eq. 3, and we set the propagation constant by taking the mean optimal value ($\alpha = 0.64$) across all interactomes.

**Network performance rankings.** For each individual gene set, we ranked the performance of all interactomes assessed with that gene set. Because not all interactomes had sufficient coverage of all gene sets, the number of interactomes assessed with each gene set varied (Fig. EV3A). To account for this, we transformed the rank values into a centralized rank. We ranked the performance scores of all $m$ successfully evaluated interactomes for each gene set and centralized this rank to give values evenly distributed around zero. Thus, for a given gene set:

$$\text{If } ranks = [1, 2, 3, …, m], \text{then } ranks_{central} = ranks - \frac{m+1}{2} \quad (5)$$

The overall rank of an interactome was calculated as the mean of the centralized ranks it achieved. As is typical for rank measures, the lower the centralized rank, the better the relative performance of an interactome. In this way, a negative centralized rank reflects an interactome in the top 50% of interactomes assessed, and a positive centralized rank reflects an interactome in the bottom 50% of interactomes assessed. This measure ensures the most weight is given to results for gene sets assessed with all interactomes while still utilizing results from smaller gene sets that could only be assessed with a minority of large networks. Centralized ranks were calculated separately for Literature, Genetic, and Experimental gene set sources for both raw and size-adjusted performance scores.

**Gene set coverage and interactome size.** We defined a subset of interactomes and gene sets to assess the influence of gene set coverage on the observed correlation between interactome size and gene set recovery performance.

1. Define a set of 16 interactomes (BioPlex 293T, BioGRID, ConsensusPathDB, DIP, FunCoup, GIANT, HPRD, HumanNet, InBio, InnateDB, MatrixDB, PTMCode2, PrePPI, ReactomeFI, SignaLink, STRING) to represent the range of interactome sizes.
2. From the previously analyzed Literature and Genetic gene sets ("original sets"), identify the maximal subsets ("reduced sets") of each gene set such that all members of each maximal subset are present in all 16 interactomes.
3. Filter the reduced sets to those containing a minimum of 30 genes for Literature sets ($n = 343$) or 15 genes for Genetic sets ($n = 46$).
4. Using the reduced sets, assess the gene set recovery performance for the 16 interactomes using the procedure outlined above (see the "Network propagation with subsampling" section) with the mean optimal gene set recovery parameters of $\rho = 0.3$ and $\alpha = 0.64$. All other genes from the original sets not present in the reduced sets are excluded from the calculation of the performance metrics to avoid incorrect labeling of true gene set genes excluded from the maximal subset as false positives.
5. For the reduced gene sets and matched original gene sets, fit linear regressions between the mean performance score and the $\log_{10}$ interaction count.
6. Calculate the size-adjusted performance for original and reduced sets separately, and extract the slopes of the fitted models for each gene set.
7. Compare the slopes between the original and reduced sets using a Wilcoxon paired test as implemented in scipy (Virtanen et al, 2020) or a sign rank test as implemented in statsmodels (Seabold and Perktold, 2010).

The Wilcoxon paired test assumes a symmetric distribution of the differences between paired observations, while the sign rank test makes no such assumption. The distribution of differences between paired

observations for Literature gene sets was significantly skewed ($skew = 6.9$, Fisher-Pearson; $p = 4 \times 10^{-12}$), calculated by scipy's skewtest (Virtanen et al, 2020). The distribution of differences was not significantly skewed for Genetic gene sets ($skew = 1.3$, $p = 0.2$). Therefore, we implemented a sign rank test for the Literature gene set slope comparison, and the Wilcoxon paired test for the Genetic gene set slope comparison.

### Creation of composite consensus networks

First, we defined "Global Composite" networks $G_k$ based on the frequency of each interaction across all 45 interactomes. For global composite networks, we defined the network threshold $k$ as the minimum number of supporting networks for each interaction. For example, $G_{k=2}$ includes all interactions present in at least two interactomes. Formally, let $i$ be an undirected interaction, $n$ be the number of interactomes, $I_{i,j}$ indicate whether an interaction $i$ is present in interactome $j$, and $k$ be the network threshold. Then:

$$G_k \subset i \, , \, if \left( \sum_{j=1}^{n} I_{i,j} \right) \geq k \tag{6}$$

Secondly, we defined "Ranked Composite" networks $R_m^k$ based on the frequency of each interaction in the top-$k$ best-performing networks. The best-performing individual networks were selected based on the average centralized rank of size-adjusted gene set recovery performance across all Literature and Genetic gene sets. For ranked composite networks, we defined the network threshold $k$ as the number of top-performing networks considered. We defined a second threshold $m$ to represent the threshold on the number of supporting networks for each interaction. For example, $R_{m=2}^{k=8}$ includes all interactions present in at least two of the top eight interactomes. Formally, let $i$ be an undirected interaction, $n$ be the total number of interactomes sorted by rank, $I_{i,j}$ indicate whether an interaction $i$ is present in interactome $j$, $k$ be the network threshold (number of top networks considered), and $m$ be the minimum support threshold. Then:

$$R_m^k \subset i \, , \, if \left( \sum_{j=1}^{k} I_{i,j} \right) \geq m \, , \, where \, m \leq k \leq n \tag{7}$$

Gene set recovery performance was assessed across a range of network thresholds. From these results, PCNet2.0, PCNet2.1, and PCNet2.2 were selected based on the mean centralized rank across Literature and Genetic gene sets. For PCNet2.1, we additionally restricted the maximum number of interactions to two million.

### Interaction and complex prediction

**Interaction prediction algorithms**. We implemented two edge prediction algorithms to assess interaction prediction accuracy: L3 (Kovács et al, 2019) and MPS (preprint: Martini et al, 2021; Wang et al, 2023). The L3 algorithm (Kovács et al, 2019) ranks predicted interactions based on the number of paths of length three between two nodes, normalized by the degree of the intermediate nodes. Formally, it calculates a degree normalized probability of interaction based on the number of paths of length three connecting two nodes $X$ and $Y$ such that:

$$L3_{XY} = \sum_{U,V} \frac{a_{XU} a_{UV} a_{VY}}{\sqrt{k_U k_V}} \tag{8}$$

Where $k_U$ is the degree of node $U$ and $a_{XU} = 1$ if proteins $X$ and $U$ interact and zero otherwise. All possible interactions are ranked based on

decreasing L3 scores to determine the most likely predicted interactions.

The Maximum similarity and Preferential attachment Score (MPS) algorithm (preprint: Martini et al, 2021) utilizes two measures of topology to rank potential interactions based on the hypothesis that the probability of two proteins interacting is proportional to the similarity between one protein and the most similar interactors of the other protein. The maximum topological similarity between two nodes is calculated from the similarities of their neighboring nodes in the network. First, define $N(u)$ as the set of direct neighbors of node $u$ and the Jaccard Similarity between two nodes $u$ and $v$ based on the similarity of their neighbors:

$$J(u,v) = \frac{|N(u) \cap N(v)|}{|N(u) \cup N(v)|} \tag{9}$$

The maximum topological similarity between nodes $x$ and $y$ is then defined based on the maximum Jaccard similarity between any neighbor of $x$: $a \in N(x)$ and $y$, and the maximum Jaccard similarity between any neighbor of $y$: $b \in N(x)$ and $x$ such that:

$$Max_{sim}(x,y) = max_{a \in N(x)} J(a,y) + max_{b \in N(y)} J(b,x) \tag{10}$$

The preferential attachment score $P$ for two nodes $x$ and $y$ is defined as:

$$P_{xy} = k_x \times k_y \tag{11}$$

Where $k$ is the node degree. All possible interactions were ranked based on both similarity metrics, consistent with previous implementations (Wang et al, 2023).

Implementations of the L3 and MPS algorithms were sourced from GitHub: L3 v.1.0.2 (https://github.com/kpisti/L3) and MPS (https://github.com/spxuw/PPI-Prediction-Project/). We standardized the evaluation of predicted interactions for both methods rather than using built in functions. STRING and TFLink were excluded from all interaction prediction tasks due to computational cost.

**Prediction of held-out self-interactions**. We measured the performance of interaction prediction for held-out self-interactions via a cross-validation procedure for each interactome, using the precision at k (P@k) as the primary evaluation metric. Our evaluation thus focused on the top predicted interactions, which are typically of most interest to researchers, and allowed more efficient calculation by avoiding consideration of up to 400 M interactions for the largest networks. To create a fair benchmark between interactomes, we set k to be the size of the test set (i.e., 10% of interactome size).

1. Partition the interactome to create 10 predict-test folds, such that prediction is performed with 90% of network edges and 10% of network edges are held out for evaluation.
2. Calculate interaction prediction scores using the L3 or MPS algorithm above. Both algorithms will generate scores for all possible interactions.
3. Calculate P@k for the predictions by defining the held-out interactions as positives. For predictions made using L3, this is achieved by taking the top-k lines of the algorithm output. For predictions made using MPS, all predictions must be sorted first

to identify the top k predictions. For efficiency, we implemented a heap structure to scan the output for the top k predictions.

In some cases, network structure and edge removal led to genes with a degree of zero. Because the prediction algorithms utilize network topology, predicting interactions involving genes with no existing interactions is not possible. Therefore, any interactions involving such genes in the held-out set were excluded from performance calculations. Due to computational constraints, ConsensusPathDB, FunCoup, and GIANT were excluded from the held-out interaction evaluation, and hu.MAP, Youn, and SignaLink were excluded from held-out interaction evaluation via MPS.

**Prediction of external interaction sets.** In addition to predicting held-out interactions, we assessed the ability of networks to predict external complex and pathway interaction sets defined by CORUM (Giurgiu et al, 2019) and PANTHER (Mi and Thomas, 2009), respectively. Interaction prediction was performed for the full interactomes with the L3 and MPS algorithms, and evaluated against the external sets of complex and pathway interactions. Any complex and pathway interactions already present in the interactomes were excluded at the evaluation step, as were any interactions whose prediction was not possible due to the absence of necessary genes. Due to computational constraints, hu.MAP was excluded from external interaction evaluation, and GIANT, ConsensusPathDB, GIANT, FunCoup, SignaLink, and Youn were excluded from external interaction prediction via MPS.

A small number of interactomes utilized CORUM and PANTHER resources for network construction and were therefore excluded from the impacted external evaluations. ReactomeFI and Pathway Commons interactomes were excluded from both CORUM and PANTHER analyses, and GeneMANIA, ConsensusPathDB, and iRefIndex were excluded from analysis with CORUM. We calculated the percent overlap of CORUM and PANTHER by assessing the proportion of the external interaction sets present in each interactome.

Finally, we defined the set of interactomes as an additional external set. We took the top 100 predictions from analysis with the full interactomes and assessed the support for these interactions based on their coverage within the corpus of other interactomes. Those not appearing in any of the 45 interactomes (network coverage of zero) were considered previously unreported.

**Prediction and evaluation of hierarchical assemblies.** Hierarchical community structures were generated using the Hierarchical community Decoder Framework (Zheng et al, 2021) (HiDeF) via the Python package hidef v1.1.5. All interactomes were assessed with a maximum resolution of 10, and all other parameters default.

- A recovered CORUM complex was defined as one reaching a Jaccard similarity of at least 0.5 with any predicted assembly.
- The GO Score was calculated for assemblies with fewer 200 genes based on the semantic similarity of genes present within each assembly (Fossati et al, 2021). The pairwise semantic similarity, implemented by goatools (Klopfenstein et al, 2018), was calculated separately for each branch of GO for each pair $u$, $v$ of genes within the assembly $N$ according to Eq. 12.

$$GO(N) = \frac{1}{3} \sum_{u \in N} \sum_{v \in N, v > u} (BP(u, v) + MF(u, v) + CC(u, v)) \tag{12}$$

**In silico assessment of predicted interactions.** Computational modeling of predicted interactions was performed using AlphaFold-Multimer (preprint: Evans et al, 2021) (AF) using ColabFold (Mirdita et al, 2022) v1.5.5 (https://pypi.org/project/colabfold/) via localcolabfold (https://github.com/YoshitakaMo/localcolabfold). For all analyses, interactions were first subset to those between genes that could be mapped to a protein sequence in UniProt (UniProt Consortium, 2023). Protein sequences were sourced from UniProt on February 5, 2024. For each protein pair:

1. Create a FASTA file containing the protein pair identifier in the definition line (e.g., ">ProteinA_ProteinB"), followed by the two protein sequences concatenated with a colon.
2. Run the localcolabfold command colabfold_batch with parameters –num-recycle 3 –model-type alphafold2_multimer_v3, and all other parameters set to defaults.
3. Extract the predicted template modeling (pTM) and interface predicted template modeling (ipTM) scores. These can be found in the "ProteinA_ProteinB_scores_rank*.json" files generated by AF.
4. For each protein pair, five models are generated by AF by default. From the pTM and ipTM scores, take the results from the model with the highest confidence (preprint: Evans et al, 2021) defined by:

$$model\ confidence = 0.8 \cdot ipTM + 0.2 \cdot pTM \tag{13}$$

For assessment with AF, we first established a baseline relationship between the network coverage of an interactome and its ipTM score. To construct this baseline, we randomly sampled groups of 50 interactions with network coverage between 1–33 interactomes and 50 interactions with network coverage ≥34.

Secondly, we assessed the ipTM scores for reported interactions within each interactome. We evaluated a sample of 50 protein pairs from each network and compared them to a background distribution of ipTM scores for 1779 randomly generated protein pairs selected independently of this study. The ipTM scores were non-normally distributed and both the network-specific and randomly generated scores displayed similar positive skew ($skew_{network} = 22$, $skew_{random} = 20$; Fisher-Pearson). Therefore, statistical testing was performed using a two-sided Mann–Whitney U-test.

Lastly, we extracted the top 50 protein pairs predicted by MPS for each interactome, requiring that both proteins had available protein structures from UniProt and that the pair was not reported as an interaction in any of the 45 interactomes studied. We defined these pairs as 'previously unreported' interactions. We modeled the interactions between these protein pairs and compared the resulting ipTM scores to the background distribution from randomly generated protein pairs using a Mann–Whitney U-test. The distribution of ipTM scores for previously unreported interactions had a similar shape to the randomly generated scores ($skew_{unreported} = 20$, Fisher-Pearson). We defined an AF-supported interaction as an ipTM score >0.53, corresponding to the 95th percentile of the background distribution.

To test whether the AF-supported classification was independent of the presence of PDB structures, we performed a $\chi^2$ independence test based on the number of AF-supported and AF-unsupported interactions with PDB structures. Because the expected number of AF-supported interactions with both proteins having PDB structures was <5, we grouped the PDB classifications of "one individual

protein," "both individual proteins," and "interaction" into a single category of "any PDB structure."

All AlphaFold-Multimer evaluations were conducted using NVIDIA Tesla V100 GPUs operating under Rocky Linux release 8.9. Each evaluation was allocated one CPU and 14 GB of memory and was subject to a 5-h time limit. Of all protein pairs, 95% of network coverage pairs, 94% of network-specific pairs, and 90% of previously unreported pairs were successfully evaluated.

## Data availability

Data presented in the figures are available in Datasets EV1–6, and full details of source databases utilized are provided in Appendix Table S1. Computer code and network resources produced in this study are available in the following databases: Computer scripts for data processing, analysis, and visualization: GitHub (https://github.com/sarah-n-wright/Network_Evaluation_Tools/releases/tag/v0.2.2). All networks utilized in this study: NDEx (https://www.ndexbio.org/index.html#/user/bae4da70-e22d-11ee-9621-005056ae23aa)—Standardized source networks: NDEx (https://doi.org/10.18119/N95C9J), PCNet2.0 network: NDEx (https://doi.org/10.18119/N9JP5J), PCNet2.1 network: NDEx (https://doi.org/10.18119/N9DW40), and PCNet2.2 network: NDEx (https://doi.org/10.18119/N9960N).

The source data of this paper are collected in the following database record: biostudies:S-SCDT-10_1038-S44320-024-00077-y.

## Peer review information

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

## Acknowledgements

This work was supported by the National Institutes of Health under awards U24 HG012107 (NHGRI), U24 CA269436 (NCI), P41 GM103504 (NIGMS), and P50 DA037844 (NIDA). The Genotype-Tissue Expression (GTEx) Project was supported by the Common Fund of the Office of the Director of the National Institutes of Health, and by NCI, NHGRI, NHLBI, NIDA, NIMH, and NINDS. The data used for the analyses described in this manuscript were obtained from the GTEx Portal on 07/11/2023. We would also like to acknowledge all of the labs and consortia involved in the development and maintenance of all the interactomes we used in this study: Agile Protein Interactomes DataServer (APID), Biomolecular Interaction Network Database (BIND), Biological General Repository for Interaction Datasets (BioGRID), Biophysical Interactions of ORFeome-based comPLEXes network (BioPlex), Compartmentalized Protein-Protein Interaction Database (ComPPI), ConsensusPathDB, Database of Interacting Proteins (DIP), FunCoup, GeneMANIA, Genome-scale Integrated Analysis of gene Networks in Tissues (GIANT, now HumanBase), High-quality INTeractomes (HINT), Human Integrated Protein-Protein Interaction rEference (HIPPIE), Human Protein Reference Database (HPRD), HumanNet, Human Protein Complex MAP (hu.MAP), The Human Reference Interactome (HuRI, formerly the Human Interactome Project), Integrated Interactions Database (IID), ZS (InBioMap, formerly Intomics), iRefIndex, InnateDB, IntAct, MatrixDB, Mentha, Molecular INTeraction database (MINT), MultiNet, Pathway Commons, Pathway Interaction Database (PID), Human Protein-Protein Interaction Prediction (PIPs), PhosphoSitePlus®, Predicted Protein-Protein Interactions (PrePPI), PROPER, ProteomeHD, PTMCode2, Reactome, SignaLink, The SIGnaling Network Open Resource (SIGNOR), SPIKE, Search Tool for Recurring Instances of Neighboring Genes (STRING), TFLink, the International Molecular Exchange (IMEx) consortium, and the Network Data Exchange (NDEx).

## Author contributions

**Sarah N Wright**: Conceptualization; Data curation; Software; Formal analysis; Validation; Investigation; Visualization; Methodology; Writing—original draft; Writing—review and editing. **Scott Colton**: Data curation; Investigation.
**Leah V Schaffer**: Software; Formal analysis; Investigation.
**Rudolf T Pillich**: Resources; Data curation; Writing—original draft.

**Christopher Churas**: Resources; Software. **Dexter Pratt**: Conceptualization; Funding acquisition; Methodology; Writing—original draft. **Trey Ideker**: Conceptualization; Resources; Supervision; Funding acquisition; Writing—original draft; Writing—review and editing.

Source data underlying figure panels in this paper may have individual authorship assigned. Where available, figure panel/source data authorship is listed in the following database record: biostudies:S-SCDT-10_1038-S44320-024-00077-y.

## Disclosure and competing interests statement

TI is a co-founder, member of the advisory board, and has an equity interest in Data4Cure and Serinus Biosciences. TI is a consultant for and has an equity interest in Ideaya Biosciences. The terms of these arrangements have been reviewed and approved by the University of California San Diego in accordance with its conflict-of-interest policies. TI is a member of the Advisory Editorial Board of The EMBO Journal. This has no bearing on the editorial consideration of this article for publication.

# Expanded View Figures

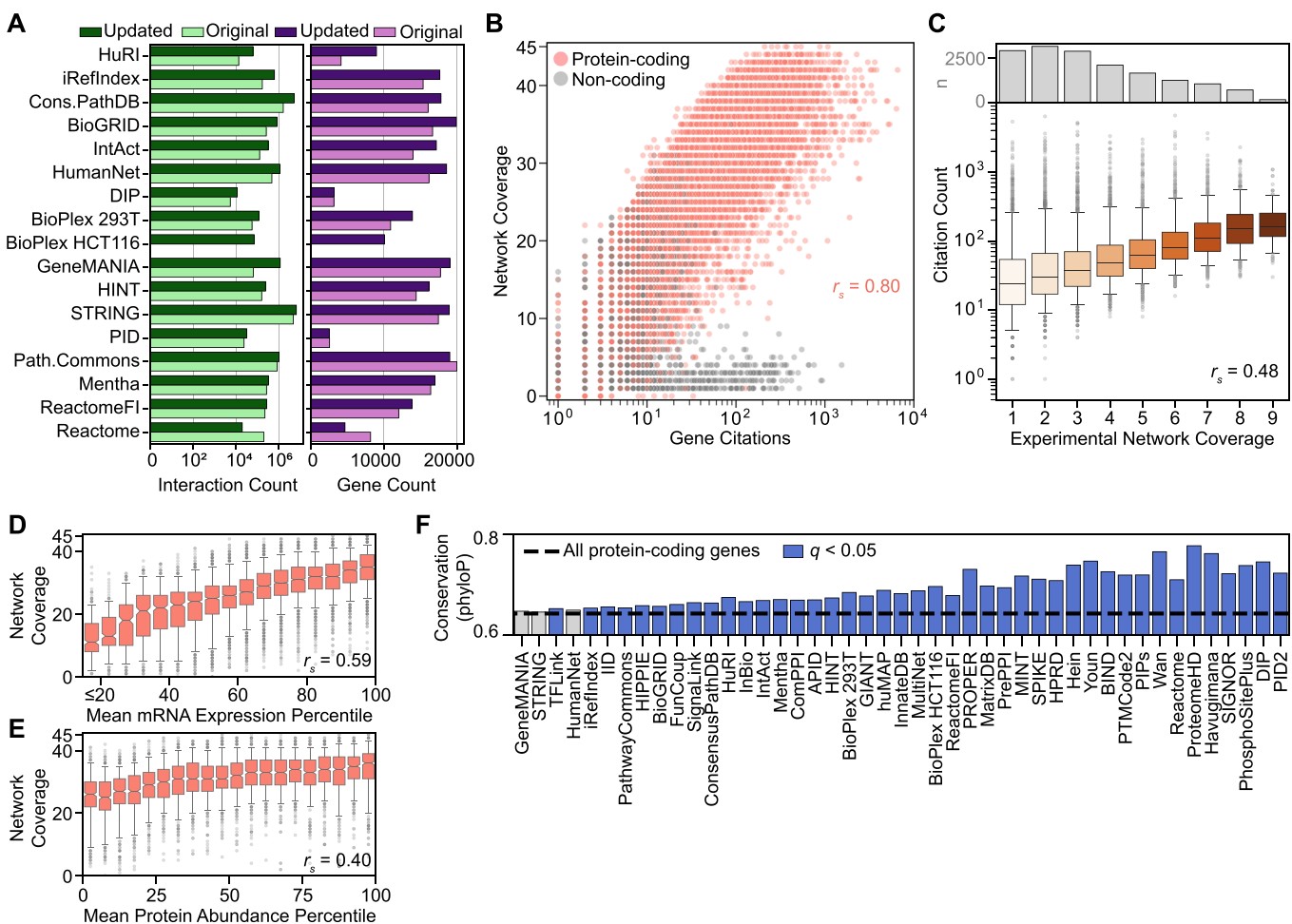

**Figure EV1. Global patterns in interactome gene representation.**

(A) Interaction and gene counts for networks updated between Huang et al (2018) and the present study. Updated networks are those where the corresponding database has been updated or where our source of the network was changed. Values represent distinct genes and interactions after data processing and mapping of identifiers to NCBI Gene IDs. (B) Comparison of network coverage and NCBI gene citation count as of December 20, 2023, for genes that are protein-coding or present in at least one interactome. Genes with no reported citations are excluded. Spearman correlation reported for protein-coding genes. (C) Box plots of citation count for protein-coding genes with at least one citation and network coverage in the nine experimental interactomes. Top bar plot shows the number of distinct interactions per network coverage value. The center of each box plot represents the median, the box boundaries correspond to the upper and lower quartiles, and the whiskers extend to the 5th and 95th percentiles. Spearman correlation reported. (D, E) Network coverage of protein-coding genes as a function of mean mRNA expression across all tissues in GTEx (D) and mean protein abundance across all tissues in the Human Protein Atlas (HPA) (E). The center of each box plot represents the median, the box boundaries correspond to the upper and lower quartiles, and the whiskers extend to the 5th and 95th percentiles. Spearman correlations reported. In (D), each box represents 1740 genes, except for the ≤20th percentile box which represents 6963 genes. In (E), each box represents between 520 and 596 genes (median 557 genes). (F) Median gene conservation scores for protein-coding genes in each interactome, compared to the median of all protein-coding genes (black line). Colored bars show networks with significantly higher median conservation scores than the baseline of all protein-coding genes (permutation test, $q < 0.05$, Bonferroni correction). Networks plotted in the same order as in Fig. 2C.

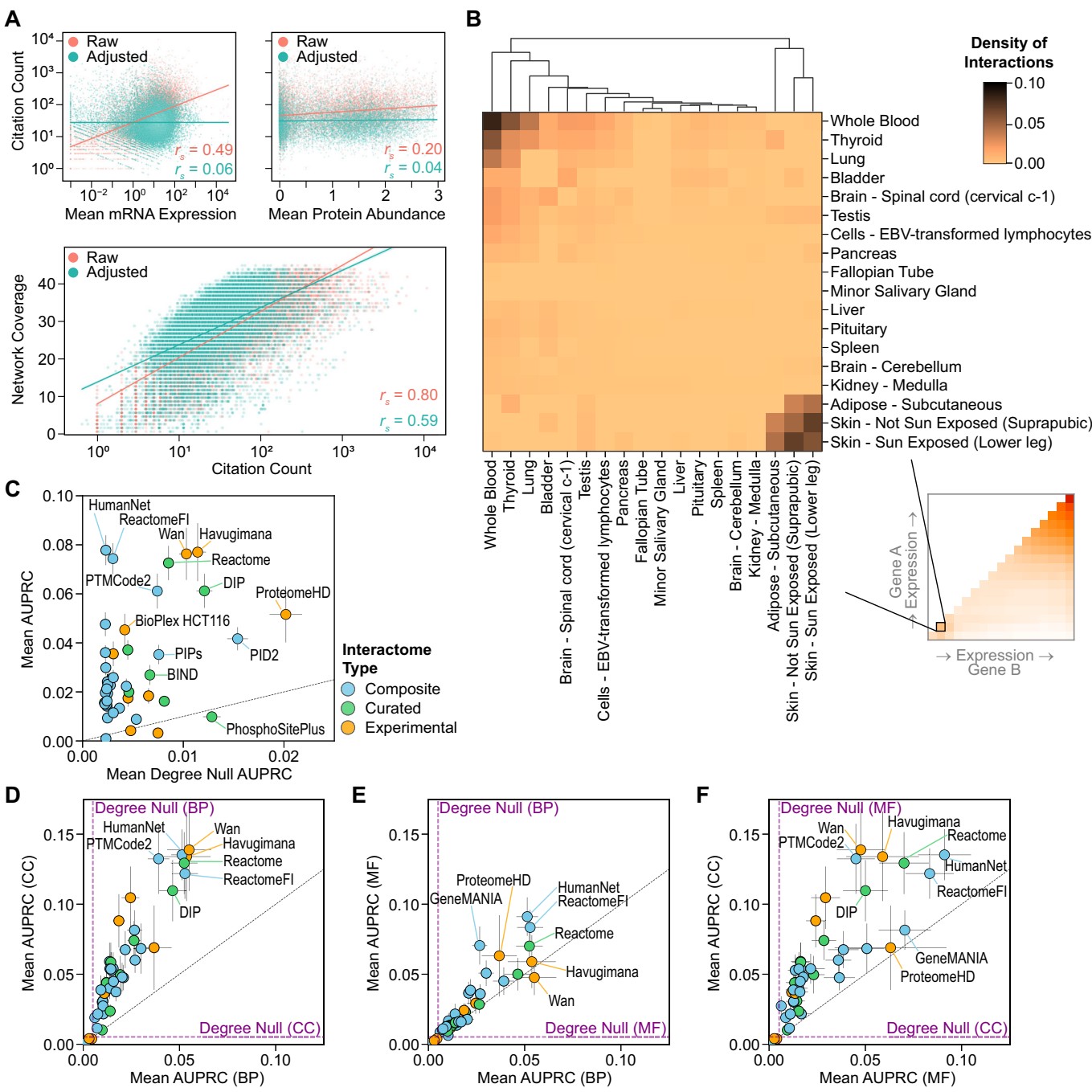

**Figure EV2. Expanded analysis of interactome biases and physiological gene function predictions.**

(**A**) Correlations between citation count and mRNA expression, protein abundance and network coverage for protein-coding genes. Original citation counts ("Raw"), and citation counts adjusted for mean gene mRNA expression using log–log ordinary least squares regression ("Adjusted"). Spearman correlations reported. (**B**) Interaction density of genes from the 20–25th percentile of mean mRNA expression across select tissues. Genes were filtered to those with non-zero expression in a maximum of two tissues (1400/1740 genes). Tissues with at least one reported interaction amongst the filtered genes are shown and clustered by minimum distance. The inset shows the corresponding bin location in Fig. 2F. (**C–F**) Gene Ontology (GO) gene function prediction via neighbor-voting. The area under the precision-recall curve (AUPRC) was calculated using 5-fold cross-validation with error bars indicating 95% confidence intervals, with mean taken across all GO annotations tested. Points colored by network classification. (**C**) Mean AUPRC across all GO annotations tested compared to mean Null AUPRC calculated from gene degree alone. The specific number of GO terms evaluated per network ranged from 268 to 1459 (median 1301) terms. See Dataset EV1 for full details. (**D–F**) Mean AUPRC for GO annotations within each GO branch (BP: biological process, CC: cellular component, MF: molecular function). Purple dashed lines show the mean null AUPRC from node degree alone across all interactomes. Black dashed lines are the identity lines. The specific number of GO terms evaluated per network per branch ranged from BP: 133 to 843 (median 731) terms, CC: 71 to 317 (median 295) terms, MF: 64 to 301 (median 278) terms. See Dataset EV1 for full details.

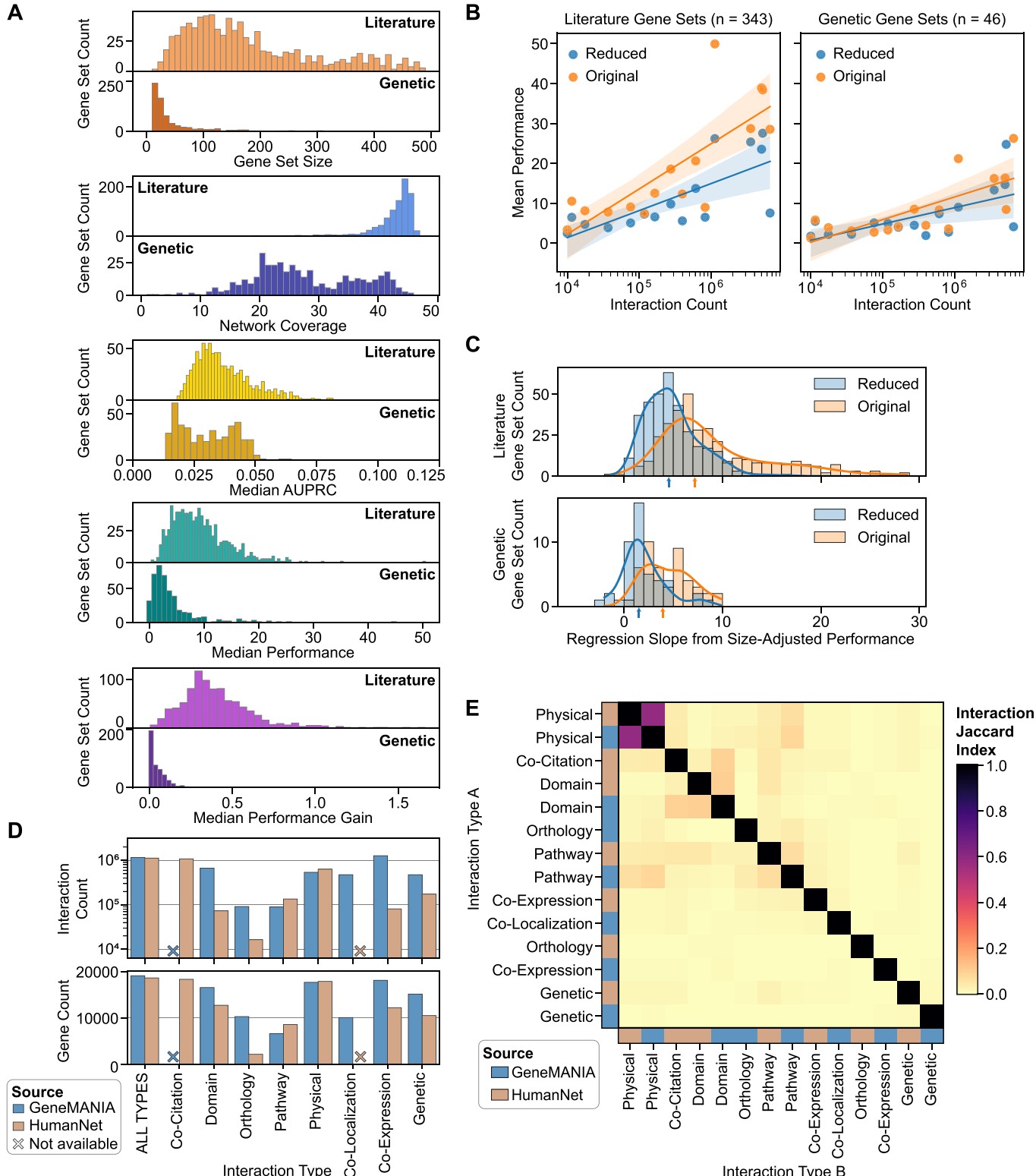

◄ **Figure EV3.   Descriptive statistics of gene set recovery performance and type-specific networks.**

(A) Distributions of gene set statistics and gene set recovery performance metrics for 45 interactomes. Gene set size: number of unique genes in each set after mapping to NCBI Gene IDs. Network Coverage: number of networks containing with at least 20 set genes. AUPRC, Performance, Performance Gain metrics: median metric of a gene set across all networks containing at least 20 set genes. See Dataset EV2 for full gene set recovery results and Dataset EV5 for full gene lists. (B, C) Gene set recovery performance for a subset of 16 interactomes, controlling for gene set coverage. Given a set of disease-associated genes, the "original" set represents the genes present in each interactome independently, while the "reduced" set represents the maximal subset of genes present in all 16 interactomes. (B) Mean gene set recovery performance with Literature and Genetic gene sets for each interactome relative to interactome size (interaction count). The lines show log-linear fits with 95% confidence intervals. See also Dataset EV2. (C) Distribution of regression slopes between interactome size and gene set recovery performance for Literature and Genetic gene sets, as calculated for size-adjusted performance. Arrows indicate the median regression slopes. (D) Sizes of interaction-type-specific interactomes defined from HumanNet and GeneMANIA. Crosses indicate that a network could not be defined from available data. Interaction type 'ALL TYPES' refers to the HumanNet and GeneMANIA networks used in the primary analysis, which include all types of interactions. (E) Interaction similarities of interaction-type-specific networks. Similarities measured by the Jaccard Index of network interactions, and clustered using the Ward variance minimization algorithm.

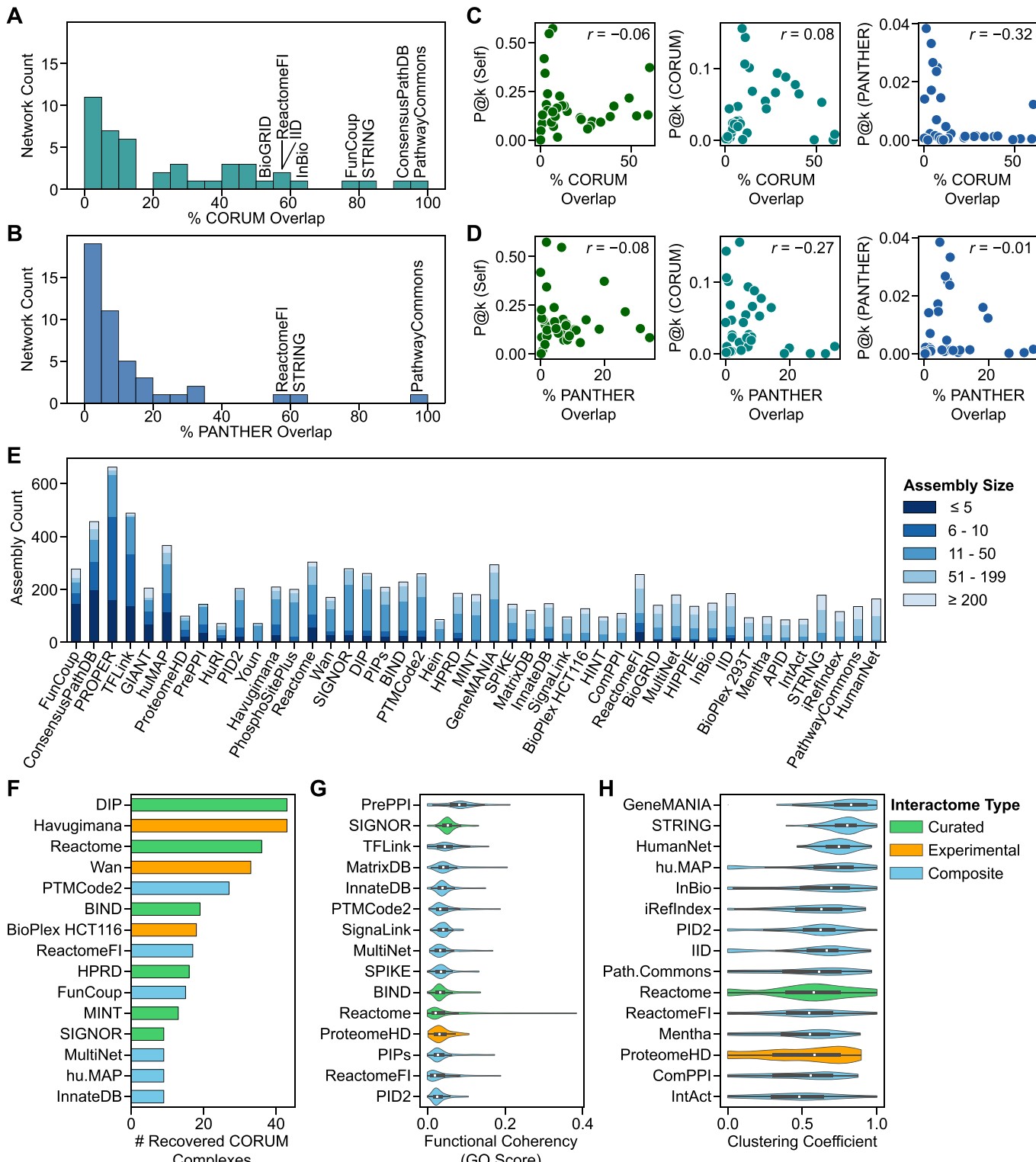

**Figure EV4. Overlap with external complex and pathway interaction sets and quality of predicted complexes.**

(A, B) Distribution of the fraction of CORUM (A) and PANTHER (B) interactions present in each interactome. Interactomes with the highest overlap are labeled. (C, D) Interaction prediction performance (mean of P@k using the MPS algorithm) as a function of interaction overlap with CORUM (C) and PANTHER (D), with associated Pearson's correlation value. (E) Size distributions of protein assemblies detected via hierarchical community detection for each interactome, sorted by median assembly size. (F–H) Evaluation of protein assemblies predicted by hierarchical community detection. The 15 top-performing networks are shown for each metric and colored by interactome classification. (F) Number of CORUM complexes recovered, defined as the number of CORUM complexes with a Jaccard similarity of ≥0.5 with any predicted assembly. (G) Functional coherency of predicted assemblies with fewer than 200 proteins based on the mean semantic similarity of GO annotations between assembly proteins (Methods). (H) Distribution of clustering coefficients of predicted assemblies with fewer than 200 proteins. In (G, H), the violins extend to the minimum and maximum observations, and represent the number of complexes with fewer than 200 genes per network as displayed in (E). The center bar represents the median, the box represents the interquartile range (Q1–Q3), and the upper and lower whiskers represent $Q1 - 1.5IQR$ and $Q3 + 1.5IQR$.

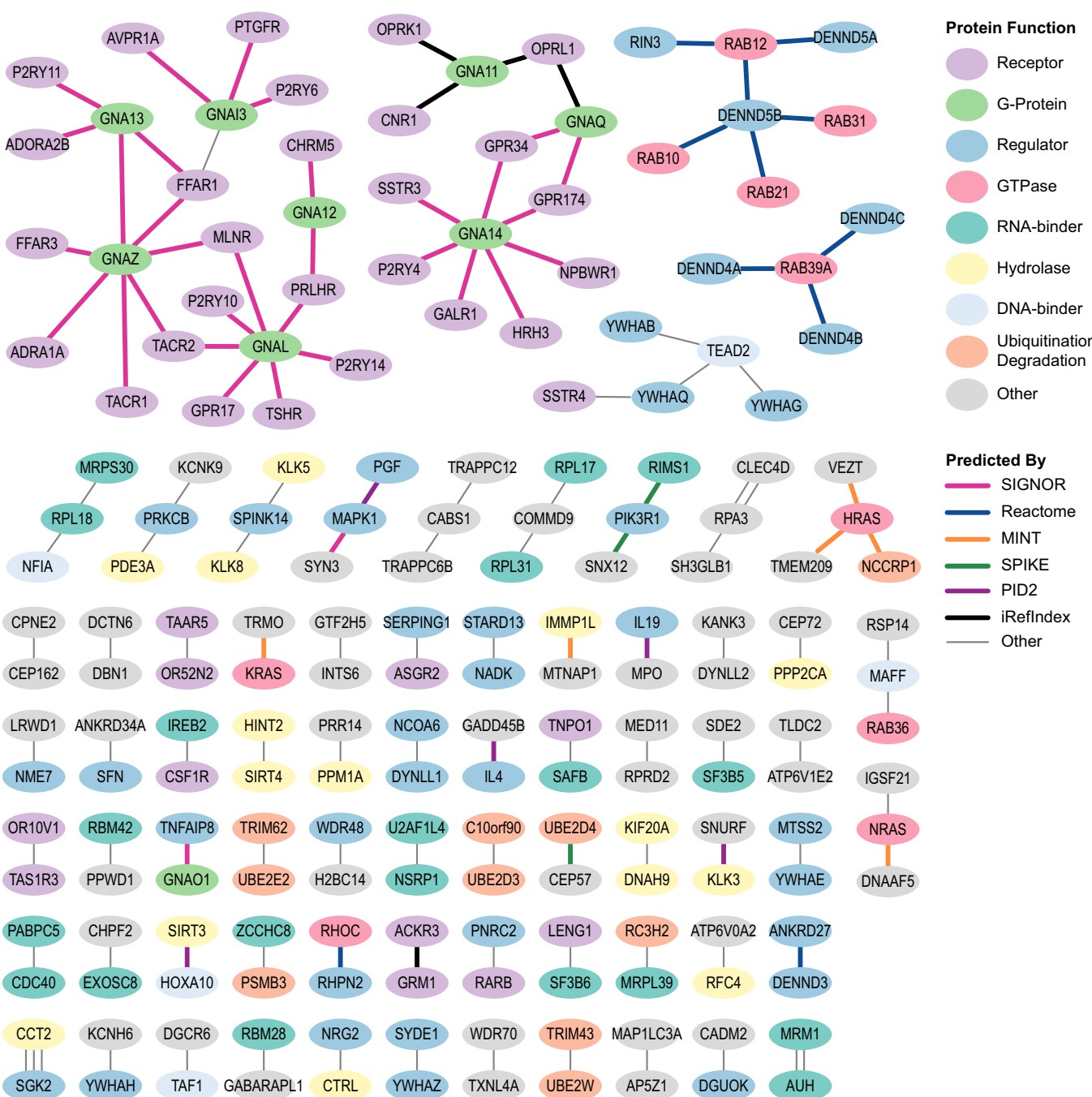

**Figure EV5.   Subnetwork of all previously unreported interactions classified as AF-supported.**

Node color indicates broad protein function and edge color indicates the predicting interactome. Protein pairs linked by multiple edges were predicted in the top 50 previously unreported interactions by multiple networks.

