## [Peer Review File · Molecular Systems Biology]

State of the Interactomes: an evaluation of molecular networks for generating biological insights

Sarah Wright, Scott Colton, Leah Schaffer, Rudolf Pillich, Christopher Churas, Dexter Pratt, and Trey Ideker

Corresponding author(s): Trey Ideker (tideker@ucsd.edu)

Review Timeline:	Submission Date:	18th Sep 24
	Editorial Decision:	21st Oct 24
	Revision Received:	7th Nov 24
	Accepted:	11th Nov 24

Editor: Poonam Bheda

Transaction Report: This manuscript was transferred to Molecular Systems Biology following peer review at Review Commons.

Review #1

1. Evidence, reproducibility and clarity:

Evidence, reproducibility and clarity (Required)

The paper entitled "State of the Interactomes: an evaluation of molecular networks for generating biological insights" presents the collection of many molecular interactomes and benchmarks to evaluate their performance as data sources to predicting gene-disease association and links. The benchmarks are further divided into sub-benchmarks that use different datasets for gene-disease associations and different validation procedures for link prediction evaluation, including one based on protein structure interface predictions with Alpha Fold. The benchmarks reveal which interactome are best suited for each task.

Introduction

- the introduction nicely introduces biases and incomplete data associated with interactome networks. It could also mention false positive interactions, which are also a rationale for combining different network sources into a consensus network, with interactions retrieved in different network sources expected to be of "higher quality" (even if this could also be the result of data leakage, see <https://academic.oup.com/bioinformatics/article/38/6/1685/6500279>).
- the introduction states the importance of interactome data in network biology. It could also mention that interactome data are also widely used as biological information sources in a wide variety of computational biology approaches, including as features in many machine learning approaches. Selecting the appropriate interactome sources is also relevant in this context.

Results

1. A plethora of interaction networks...

- HumanNet interactome: I was a bit confused by the HumanNet network(s) which is sometimes referred as HumanNet v3, sometimes HumanNet-XC, and HumanNet-FN is also mentioned as included in XC. It is however not clear what XC and FN stands for, or if HumanNet v3 is the XC network? The method section detail interaction types but called "HS-CC", "HS-CX".. And XC/FN are not mentioned in the method section. This could be clarified.
- "All networks ... achieving a reduction of over 60%" => a reduction as compared to what? This sentence could also clarify what is the goal of the processing pipeline (map gene names? Remove duplicates?)
- 14/21 interactomes are updated from the previous publication of the team. For the remaining interactomes, what is the reason for not updating them? Is it because they did not evolve since last publication?
- The end of this section could also be more precise about the meaning of "interdependencies", which are referred to as "dependency" or "impact" in the figure's legend. It is not fully clear to me how this impact is computed. The legend states that it's "cited as a source", but where? In the publication associated with the interactome? The protocol could be described more clearly but, in any case, it is a very interesting idea that showcases the efforts of the different databases. The

associated Figure 1C is also very interesting.

2. Gap remains...

- I had difficulties understanding the data and the construction of the Figure 2H-I. The corresponding sentences in the text, figure's legend, and method section, are not homogeneous. For instance, the figure's legend states that "all the genes are binned based on increasing citation counts..." whereas the text focuses on "high connectivity", and connectivity is not mentioned in the legend.

- "...had more restricted interactions" not clear to me what are restricted interactions.

- The Figure 2F is quoted in the discussion but not in the main text. The legend associated is a bit unclear regarding "with at least one significantly under-enriched network", why using this filtering criteria? The discussion regarding this experiment states that "not all biological domains are equally represented", but this analysis also emphasizes some network biases, no?

3. Large composite networks...

- It is not clear how the "experimental studies" were selected from the literature; this could be clarified in the method section.

- Figure 3A: Why is a distribution drawn for the 50 iterations on the shuffle diffusion model but only the mean computed for the observed diffusion model? A distribution could also be drawn from the 50 iterations on the true gene sets, no?

- Figure 3D, clarify the meaning of "baseline" that is not used elsewhere in the manuscript.

- An important point of the benchmark to evaluate the network performances based on network propagation is the selection of the restart parameter and of the held-out/sampled seeds. The authors propose a method to optimise these parameters for each interactome. The rationale for such a strategy could be better explained and justified because one could argue that to compare the predictive power performances of the different networks, the same restart parameter and set of seeds could be used. In addition, this optimisation is done on a specific gene annotation dataset (Hallmark), but one could imagine for instance doing this optimisation using gene-disease association and then comparing performance using the hallmark annotations. Here also, I am missing some rationale. Finally, concerning the optimisation of the selection of held-out versus seed genes, I don't understand why the optimal is not always 1 seed and all-but-one held out gene nodes?

4. Not all interactions are created equal

- "...reflecting the influence of differing network construction procedures", do we have hypotheses on what are these network construction procedures and how they precisely could impact the analyses?

6. Evaluation and in silico validation

- Cross-validations are necessary to evaluate prediction performances with internal held-out interaction data but I'm not sure why cross validations are also applied to evaluation with external gold-standard interactions? Such evaluation could be done directly on novel predictions without anything held-out, no?

- I guess CORUM and PANTHER data are not included in any of the 46 initial interactomes, but are there any conformation of that?

- Figure 5D: I don't get what is an "orthogonal" network coverage?

Methods

- protein "abundance", as used in the main text, should be preferred over "protein expression", which is used in the method section.
- For protein abundance, it is not clear if the data are provided in categorical levels or quantitative values or both as categories are mentioned after the sentences about mean computation.

2. Significance:

Significance (Required)

This manuscript is a follow-up on a previous paper (Huang et al. 2018 Cell Systems) with extended interactome and benchmarks.

Molecular networks are widely used directly to study gene and protein cellular functions as well as data features in a large palette of machine learning computational biology tools. But the number of available networks has been growing steadily and it's often not easy for the users to navigate the interactome zoo and decide which network to consider. In this context, the work proposed here is expected to be very helpful to an audience of computational biologists using network data. The manuscript is clear and well-written, organised in logical steps. The figures are informative and of high quality.

3. How much time do you estimate the authors will need to complete the suggested revisions:

Estimated time to Complete Revisions (Required)

(Decision Recommendation)

Less than 1 month

4. Review Commons values the work of reviewers and encourages them to get credit for their work. Select 'Yes' below to register your reviewing activity at Web of Science Reviewer Recognition Service (formerly Publons); note that the content of your review will not be visible on Web of Science.

Yes

Review #2

1. Evidence, reproducibility and clarity:

Evidence, reproducibility and clarity (Required)

The manuscript submitted by Wright et al describe a comparative analysis of 46 human protein-protein interaction (PPI) networks that were generated from different sources being experimental, curation, or composite. The PPI networks were evaluated based on two criteria: their performance for the prediction of disease genes and for the prediction of novel PPIs. A couple more analyses were presented relating to network coverage, incorporation of networks in other resources and GO enrichments. The authors conclude that larger composite networks are better for disease gene prediction while smaller signaling pathway-related networks are better for PPI prediction.

While the presented analyses are mostly thorough the focus on disease gene and PPI prediction seem very restrictive given the rather general statements made in the title and abstract that these networks were evaluated for their ability to generate biological insights. Other relevant applications are for example (physiological) gene function predictions using guilt-by-association or the prediction of protein complex assemblies.

The conclusions lack a more detailed discussion, e.g. why is it that the signaling networks perform better for PPI prediction when evaluated against predictability with AlphaFold? Could there be a potential bias because these smaller signaling networks like SIGNOR, SPIKE and REACTOME appear to be high quality networks due to manual curation. I guess the authors have ensured that predicted PPIs do not have a resolved structure in the PDB yet? I wonder to which extent potential circularity in literature curation on one side and training on structures in the PDB on the other side lead to higher validation rates by AF for these types of networks. Would these signaling networks consist more of direct PPIs compared to other networks? This could also lead to higher AF modelling successes. Just to give some ideas. In the same way, a more in depth discussion and potential analysis might be warranted for why larger networks perform better for the prediction of disease genes. From the methods section I understand that disease genes were filtered to those being contained in a respective network. Did the authors thoroughly check for any other imbalance that might arise because of large differences in network sizes and how this impacts the gene sets used to evaluate the networks?

The use of AlphaFold for its ability to predict structural models of PPIs is a reasonable complementary approach to gain confidence in PPI predictions but is overstated in its ability to almost replace experimental validation. The use validation in this context is over-optimistic. To the best of my understanding, no clear performance evaluations exist of AF for its ability to predict PPIs. I suggest more carefully rephrasing the corresponding results section. Given different performances of the networks for PPI prediction as evaluated by AF, it is also rather unclear what the specific recommendations of the authors are with respect to which networks should be used for PPI prediction.

The methods section could provide at places some more detail, i.e. how do the PPI prediction algorithms L3 and MPS(T) work? What is the similarity by which proteins are assessed for the MPS? It is also not clear what the authors mean by random protein pairs. At places this could mean a random selection of PPIs from a network, at other places this could mean a generation of random pairs of proteins. This should be better specified.

The results section could be improved in clarity if clear hypotheses were formulated motivating the choice for the analyses presented.

Figure 2F is referred to as Figure 2I. Please check the results and discussion section.

2. Significance:

Significance (Required)

In general, it remains unclear how the findings reported in the study advance the field in understanding which of the networks are recommended for the use of which application. Biases in the networks arising from literature curation, expression levels, and gene conservation are not clearly quantified but this would be important to understand which networks should be used for which application, also to avoid circularity for some applications.

This study would be of potential interest to a specialized group of computational researchers interested in the use of PPI networks for predicting biological function.

3. How much time do you estimate the authors will need to complete the suggested revisions:

Estimated time to Complete Revisions (Required)

(Decision Recommendation)

Between 1 and 3 months

4. Review Commons values the work of reviewers and encourages them to get credit for their work. Select 'Yes' below to register your reviewing activity at Web of Science Reviewer Recognition Service (formerly Publons); note that the content of your review will not be visible on Web of Science.

No

Full Revision

Manuscript number: RC-2024-02501

Corresponding author(s): Trey Ideker

1. Point-by-point description of the revisions

This section is mandatory. Please insert a point-by-point reply describing the revisions that were already carried out and included in the transferred manuscript.

Reviewer #1

Evidence, reproducibility and clarity

The paper entitled "State of the Interactomes: an evaluation of molecular networks for generating biological insights" presents the collection of many molecular interactomes and benchmarks to evaluate their performance as data sources to predicting gene-disease association and links. The benchmarks are further divided into sub-benchmarks that use different datasets for gene-disease associations and different validation procedures for link prediction evaluation, including one based on protein structure interface predictions with Alpha Fold. The benchmarks reveal which interactome are best suited for each task.

Introduction

- the introduction nicely introduces biases and incomplete data associated with interactome networks. It could also mention false positive interactions, which are also a rationale for combining different network sources into a consensus network, with interactions retrieved in different network sources expected to be of "higher quality" (even if this could also be the result of data leakage, see <https://academic.oup.com/bioinformatics/article/38/6/1685/6500279>).

We have added this important context alongside our comments on network incompleteness in the **Introduction**:

"Many factors could potentially cause skew, including experimental constraints or biases towards highly studied or highly expressed genes¹⁵. Beyond gaps in interactome knowledge, network resources may contain false positive interactions, which can dilute biological signals. Understanding these biases and other effects is important, as any bias in an interactome will likely be reflected in analyses that use that interactome."

And to our motivation for **Consensus networks improve gene set recovery performance**:

"Such an approach leverages the complementary information contained in different databases while excluding non-reproduced interactions."

The issue of data leakage is particularly relevant to our parsimonious composite networks (PCNets), and we have expanded on this point in the **Discussion**:

"In contrast, some high-performing interactomes' rankings suffered after size adjustment, indicating that they may contain many lower-quality interactions that are nevertheless overcome by the network propagation procedure. We found that some low-quality and potentially spurious interactions can be excluded by selecting only those interactions present in multiple sources. In this vein, we used the size-adjusted interactome rankings to construct

high-quality parsimonious composite networks (PCNets), requiring all interactions to be supported by at least two databases. Despite our approach not explicitly accounting for redundancy between sources⁶⁴, the higher gene set recovery performance (Figure 4) indicates an improvement in true biological signal.”

- the introduction states the importance of interactome data in network biology. It could also mention that interactome data are also widely used as biological information sources in a wide variety of computational biology approaches, including as features in many machine learning approaches. Selecting the appropriate interactome sources is also relevant in this context.

We agree that networks are widely used in many computational biology approaches. We felt this point fit well in the **Discussion** and have added it as follows:

“Additionally, interactome data are increasingly used to provide machine learning features and determine the architecture of neural networks. These applications warrant a specific and detailed analysis of interactome resources beyond our study.”

Results

1. A plethora of interaction networks...

- HumanNet interactome: I was a bit confused by the HumanNet network(s) which is sometimes referred as HumanNet v3, sometimes HumanNet-XC, and HumanNet-FN is also mentioned as included in XC. It is however not clear what XC and FN stands for, or if HumanNet v3 is the XC network? The method section detail interaction types but called "HS-CC", "HS-CX" .. And XC/FN are not mentioned in the method section. This could be clarified.

We have clarified references to HumanNet networks throughout the text and figures. To reduce confusion, we now define the version containing all interaction sources as a single 'HumanNet' network in our primary analyses. Correspondingly, the number of interactomes in our corpus is now 45. The cocitation-free version is now utilized only in Figure 4, consistent with the usage of cocitation-free STRING, and now termed 'HumanNet (CC-free).'

In addition, we have modified **Methods: Data Collection and Processing - Collection and Standardization of interaction data** and **Creation of interaction-type-specific networks** to clearly link all HumanNet interactomes used in the manuscript to the terminology used at <https://staging2.inetbio.org/humannetv3/>:

“We used the functional network extended by co-citation (HumanNet-XC) as the primary HumanNet interactome and defined the functional subnetwork (HumanNet-FN) as the cocitation-free HumanNet (CC-free)...

From HumanNet, we downloaded HumanNet-co-citation (HS-CC), HumanNet-coexpression (HS-CX), HumanNet-pathway (HS-DB), HumanNet-domain (HS-DP), HumanNet-genetic interaction (HS-GI), HumanNet-phylogenetic similarity (HS-PG), and HumanNet-physical (HS-PI). These type-specific HumanNet networks were constructed via combinations of curated and experimental sources under a supervised learning framework. All networks were downloaded from <https://staging2.inetbio.org/humannetv3/> and standardized per our pipeline outlined above...

While drawing on different data types, we assigned HumanNet-phylogenetic similarity and GeneMANIA-predicted to the category 'Orthology' as both utilized information from non-human species.”

- "All networks ... achieving a reduction of over 60%" => a reduction as compared to what? This sentence could also clarify what is the goal of the processing pipeline (map gene names? Remove duplicates?)

For improved clarity, we have updated the ***A plethora of interaction networks from diverse but overlapping sources*** to read:

"All networks were standardized to map gene identifiers to NCBI Gene IDs, remove duplicates and self-interactions, and filter to human interactions where applicable."

We have also moved the identifier-mapping statistic to a more logical position in ***Methods: Data Collection and Processing - Standardization of gene identifiers*** alongside the details of the identifier-mapping procedure.

"The performance of our gene mapping pipeline was compared to the use of MyGeneInfo alone, achieving a reduction of over 60% in genes that could not be mapped to NCBI gene identifiers (Figure S9A)"

- 14/21 interactomes are updated from the previous publication of the team. For the remaining interactomes, what is the reason for not updating them? Is it because they did not evolve since last publication?

We have clarified this point as follows in ***A plethora of interaction networks from diverse but overlapping sources***:

"Building on the 21 networks analyzed in our prior publication¹⁶, we noted that 14 of these had been significantly updated while 7 represented static databases; we also extended the corpus with 24 additional interactomes."

- The end of this section could also be more precise about the meaning of "interdependencies", which are referred to as "dependency" or "impact" in the figure's legend. It is not fully clear to me how this impact is computed. The legend states that it's "cited as a source", but where? In the publication associated with the interactome? The protocol could be described more clearly but, in any case, it is a very interesting idea that showcases the efforts of the different databases. The associated Figure 1C is also very interesting.

We agree that this point is unclear and have harmonized the terminology in the results ***A plethora of interaction networks from diverse but overlapping sources***:

"While the number of distinct databases has increased substantially, most available networks rely on similar sources and have extensive dependencies (Figure 1C). The interactomes most frequently identified as dependencies of other network resources were BioGRID²⁶ (19/45), IntAct²⁷ (20/45), DIP²⁸ (16/45), MINT²⁹ (15/45), and HPRD³⁰⁻³² (15/45)."

We have also updated the **Figure 1C** legend:

"Node size indicates the "Dependency Count," defined as the number of times a given interactome was identified as a source by other evaluated network sources (Methods)."

And have added additional details to ***Methods: Data Collection and Processing - Collation of network metadata***.

“Network dependencies and interaction types were collated from publicly available information based on definitions in Supplemental Table 6. Dependencies and interaction types were manually curated from associated publications and websites, as well as metadata available within the interactome data files (such as ‘Source’ or ‘Database’ columns). Where available, PMIDs associated with interactions were used to identify experimental sources. A network dependency between a target and a source network was defined as a relationship where interactions from the source network were directly incorporated into the target network. Using a network to train, prioritize, or score interactions was not considered a dependency.”

2. Gap remains...

- I had difficulties understanding the data and the construction of the Figure 2H-I. The corresponding sentences in the text, figure's legend, and method section, are not homogeneous. For instance, the figure's legend states that "all the genes are binned based on increasing citation counts..." whereas the text focuses on "high connectivity", and connectivity is not mentioned in the legend.

We have harmonized all sections and the corresponding figures to use the term ‘interaction density,’ which more clearly communicates these findings. Please note the original Figures 2H-I are now Figures 2F-G.

We have clarified the terminology in ***Gaps remain in coverage of the human proteome:***

“High expression and citation levels were generally associated with high interaction density (Figure 2F). For example, among the 247 most cited genes, 80% of possible pairwise interactions were reported at least once, while only 0.1% of possible interactions were reported between the 480 least cited genes. However, an increased density of interactions was observed among a subset of genes with low mean mRNA expression - driven by interactions between tissue-specific genes of whole blood, thyroid, and skin (Figure S4B). In contrast, the interaction density was largely uniform within and between chromosomes, except for the Y and mitochondrial chromosomes (Figure 2F).”

We have also updated **Figure 2**, and the corresponding caption:

“The interaction density is calculated per pair of bins as the number of observed interactions divided by the number of possible interactions between all pairs of gene A and gene B for a bin pair. The diagonal entries, therefore, represent the interaction density between genes with similar annotation values, while off-diagonal entries represent the interaction density between genes with differing annotation values.”

- "...had more restricted interactions" not clear to me what are restricted interactions.

We have updated this statement to use ‘interaction density’ (as above) in ***Gaps remain in coverage of the human proteome:***

“Analysis of interaction density among GO Biological Process annotations highlighted a high density of interactions among genes involved in a broad range of genomic organization and processing functions (Figure 2G). In contrast, genes associated with some functions and components, such as cyclase activity and the peroxisome, had a high density of interactions amongst themselves but few interactions with outside genes”

- The Figure 2F is quoted in the discussion but not in the main text. The legend associated is a bit unclear regarding "with at least one significantly under-enriched network", why using this filtering criteria? The discussion regarding this experiment states that "not all biological domains are equally represented", but this analysis also emphasizes some network biases, no?

The paragraph below was mistakenly omitted in the submitted manuscript file. We thank the reviewer for catching this oversight and have now included it in **Gaps remain in coverage of the human proteome**. Please note that the original Figure 2F is now Figure 2D.

"We next hypothesized that differences and biases in the genes represented in each interactome would correspond to gaps in biological function. Gene set enrichment analysis using Gene Ontology (GO) annotations showed that experimental interactomes favor translational machinery while being systematically under-enriched for receptor-ligand and cyclase activities, possibly due to experimental limitations (Figure 2D). Furthermore, many curated and small composite networks such as SIGNOR³⁹, PID v2.0³⁵, and DIP²⁸ were enriched for inflammation and immune response genes, while protein glycosylation and transporter activity were generally underrepresented."

We also agree that the results presented in Figure 2 reveal general patterns across all networks as well as biases of specific networks. In addition to the specific results presented in the above paragraph, we have modified the **Discussion** to read:

"Many of these interactomes now cover nearly all protein-coding genes (Figure 2A), though not all biological domains are equally represented, with biases particularly prevalent amongst experimental networks (Figure 2C-D, Table 1). Consequently, biological representation should be carefully considered when selecting a network."

The filtering in Figure 2D was chosen to highlight gaps in biological representation (under-enrichment) while maintaining a reasonable figure size. We have clarified the **Figure 2** caption:

"To highlight GO terms that are poorly represented in at least one interactome, the visualization includes networks with < 10,000 genes and terms with at least one significantly under-enriched network ($q < 0.01$). See Supplemental Table 2 for the full results."

3. Large composite networks...

- It is not clear how the "experimental studies" were selected from the literature; this could be clarified in the method section.

We have expanded on the selection criteria in the **Methods: Data Collection and Processing - Definition of gene and interaction sets**:

"Lastly, we identified gene sets generated from experimental data published between January 1 and January 24, 2024. Candidate studies were identified from PubMed using grouping keywords such as 'module,' 'profile,' 'gene set,' and 'component,' combined with experimental context keywords such as 'transcriptional,' 'regulatory,' 'differentially expressed,' 'biomarker,' 'single-cell,' and 'gene expression.' Studies were selected if they generated publicly available gene sets between 20 and 250 genes (or data from which these could be readily defined) and did not utilize literature or network resources for set definition, resulting in 17 experimental gene sets. To analyze each interactome, we selected genesets with a minimum of 20 (literature, genetic) or 10 (genetic 2023+, experimental) overlapping genes."

- Figure 3A: Why is a distribution drawn for the 50 iterations on the shuffle diffusion model but only the mean computed for the observed diffusion model? A distribution could also be drawn from the 50 iterations on the true gene sets, no?

The reviewer is correct that the sub-sampling procedure generates 50 AUPRC values, which we aggregate into a single mean AUPRC value. However, the distribution shown in gray in Figure 3A represents 50 of these mean AUPRC values. The reason is that we generated 50 randomized versions of each interactome by network shuffling and performed the gene set recovery procedure to generate a null distribution of mean AUPRC values, where each individual value represents an underlying distribution of 50 sub-sampling iterations. While Figure 3A is consistent with the statistical test performed, we have **updated the annotation in Figure 3A** to clarify the origin of the distribution:

“Null distribution of mean AUPRC values from gene set recovery (steps 2-6) with 50 shuffled interactomes.”

- Figure 3D, clarify the meaning of "baseline" that is not used elsewhere in the manuscript.

We have updated this terminology from 'Baseline' to 'ALL TYPES' to more clearly reference the full HumanNet and GeneMANIA networks which are composites of all interaction types. We have **updated Figure 3**, and to the caption we have added:

“‘ALL TYPES’ refers to the full HumanNet and GeneMANIA networks.”

- An important point of the benchmark to evaluate the network performances based on network propagation is the selection of the restart parameter and of the held-out/sampled seeds. The authors propose a method to optimise these parameters for each interactome. The rationale for such a strategy could be better explained and justified because one could argue that to compare the predictive power performances of the different networks, the same restart parameter and set of seeds could be used. In addition, this optimisation is done on a specific gene annotation dataset (Hallmark), but one could imagine for instance doing this optimisation using gene-disease association and then comparing performance using the hallmark annotations. Here also, I am missing some rationale. Finally, concerning the optimisation of the selection of held-out versus seed genes, I don't understand why the optimal is not always 1 seed and all-but-one held out gene nodes?

Firstly, we acknowledge that we could have used the same parameters for every interactome, but we chose to perform parameter optimizations in order to evaluate each interactome under its best conditions. We have elaborated on our rationale in ***Methods: Gene Set Recovery Evaluation - Parameter optimization for gene set recovery***.

“Previous studies have shown that the optimal gene set recovery parameters can vary between networks¹⁶. Therefore, to enable assessment of interactomes under their best conditions, we performed optimization of the two key gene set recovery parameters (network propagation constant) and p (subsampling proportion) for each interactome and gene set using the 50 MSigDB Hallmark pathways⁸⁶ as an independent source of gene sets. The Hallmark pathways provide a source of high-quality gene sets, independent of the larger and disease-focused literature, and genetic and experimental gene sets used for evaluation.”

Under our gene set recovery framework, we would expect using a singular seed gene to be suboptimal. A major benefit of network propagation is the ability to aggregate signals from multiple

genes and to represent the genetic architecture of complex traits where, for example, there may be multiple underlying functional pathways embedded in multiple regions of the network.

4. Not all interactions are created equal

- "...reflecting the influence of differing network construction procedures", do we have hypotheses on what are these network construction procedures and how they precisely could impact the analyses?

Differences in the underlying data and the use of a supervised learning framework by HumanNet are likely to drive these differences. We have expanded on one such example in ***Not all interaction types are created equal***:

"... reflecting the influence of differing network construction procedures. For example, the higher performing GeneMANIA-Orthology network includes interactions predicted from non-human species⁵⁰, a data source excluded in the latest version of HumanNet⁴¹."

We have also added more clarity to the construction procedures in ***Methods: Data Collection and Processing - Creation of interaction-type-specific networks***:

"These type-specific HumanNet networks were constructed via combinations of curated and experimental sources under a supervised learning framework. All HumanNet networks were downloaded from <https://staging2.inetbio.org/humannetv3/> and standardized per our pipeline outlined above...

From GeneMANIA ... All interactions within a source were concatenated to form interaction-type-specific GeneMANIA networks. GeneMANIA sources for genetic and coexpression interactions include large-scale screens with interaction scores. Therefore, based on interaction scores, we filtered these interaction-type-specific networks to the top 10% of genetic interactions and the top 3% of co-expression interactions. While drawing on different data types, we assigned HumanNet-phylogenetic similarity and GeneMANIA-predicted to the category 'Orthology' as both utilized information from non-human species."

6. Evaluation and in silico validation

- Cross-validations are necessary to evaluate prediction performances with internal held-out interaction data, but I'm not sure why they are also applied to evaluation with external gold-standard interactions. Such evaluation could be done directly on novel predictions without anything held out, no?

The reviewer is correct that for testing with an external gold standard it is not necessary to perform cross-validations. However, as outlined in the below comment, our 'gold standards' cannot be considered truly independent of all interactomes (**new Supplemental Fig6A-B**). Therefore, we have now excluded the several interactomes that incorporate CORUM and PANTHER interactions and, for the remaining networks, performed the analysis without cross-validation as suggested by the reviewer (**updated Figure 5B-C**). Further to this update, we now more accurately refer to CORUM and PANTHER as external interaction sets, rather than 'gold standards.'

This same reviewer suggestion has also impacted our analysis of network coverage (Figure 5D) and *in silico* interaction prediction using AlphaFold-Multimer (Figure 5F). These analyses now consider predictions from the entire interactomes without cross-validation (**updated Figure 5D & G, new Supplemental Figure 7D**). We have modified the text presenting these results in ***Evaluation of predicted interactions and complexes***:

“The predicted interactions were assessed against held-out interactions from the same network using a 10-fold cross-validation procedure, as well as external sets of physical interactions from multimeric protein complexes recorded in the Comprehensive Resource of Mammalian Protein Complexes (CORUM) database⁵³ and pathway interactions derived from curated, primarily signaling, pathways from the PANTHER knowledgebase⁵⁴.”

We have also updated the corresponding **Methods: Interaction Prediction - Prediction of external interaction sets**.

“Interaction prediction was performed for the full interactomes with the L3 and MPS algorithms, and evaluated against the external sets of complex and pathway interactions. Any complex and pathway interactions already present in the interactomes were excluded at the evaluation step, as were any interactions whose prediction was not possible due to the absence of necessary genes. A small number of interactomes utilized CORUM and PANTHER resources for network construction and were therefore excluded. ReactomeFI and Pathway Commons interactomes were excluded from both CORUM and PANTHER analyses, and GeneMANIA, ConsensusPathDB, and iRefIndex were excluded from analysis with CORUM. We calculated the percent overlap of CORUM and PANTHER by assessing the proportion of the external set interactions present in each interactome.”

Figure 5. Interaction prediction performance and AlphaFold evaluation. A) Interaction prediction evaluation by L3 and MPS algorithms for held-out self-interactions via 10-fold cross-validation. Mean (bar) and individual fold (points) precision at k ($P@k$, $k = \text{size of test set}$) for the 15 top-performing interactomes. B-C) Interaction prediction evaluation by L3 and MPS algorithms for external interaction sets defined from (B) CORUM complex interactions and (C) PANTHER pathway interactions for the 15 top-performing interactomes. D) Distribution of network coverage for the top 100 interactions predicted by each interactome. A network coverage of 0 indicates that an interaction was not reported in any of the 45 interactomes. The top bar plot shows the number of edges in the corresponding interactome. E) Relationship between AlphaFold-Multimer interface predicted TM-score (ipTM) and interaction network coverage. F) Interactomes enriched for interactions with high interface scores ($q < 0.15$, Mann-Whitney U-test, BH-corrected), using samples of 50 network interactions compared to a background distribution of 1779 randomly generated protein pairs. G) AlphaFold assessment of previously unreported interactions predicted using MPS. AF-supported interactions were defined as protein pairs with an interface score in the 95th percentile (ipTM > 0.56, dashed line) of the background distribution of randomly generated protein pairs. The significance of an interactome's mean ipTM score for previously unreported interactions was assessed by a Mann-Whitney U-test compared to the

distribution of scores from randomly generated protein pairs (* $q < 0.1$, *** $q < 10^{-5}$, BH-corrected). See also Supplemental Tables 4 and 5.

Supplemental Figure 7. AlphaFold-Multimer predictions and presence of experimentally resolved structures from PDB.... D) ipTM scores for previously unreported interactions predicted by each interactome using the MPS algorithm. The green line represents the 95th percentile of ipTM scores for randomly generated protein pairs, with protein pairs above this threshold defined as AF-supported interactions.

- I guess CORUM and PANTHER data are not included in any of the 46 initial interactomes, but are there any conformation of that?

We agree this is an important point to clarify in the text. In fact, a minority of interactomes directly incorporate CORUM (ReactomeFI, GeneMANIA, Pathway Commons, Consensus Path DB, iRefIndex) and/or PANTHER (ReactomeFI, Pathway Commons) and we have now excluded them from analysis with external interaction sets (as outlined in the above comment). In **new Supplemental Figure 6A-D**, we show the overlap with CORUM and PANTHER, and that the degree of overlap with the CORUM and PANTHER databases does not drive the interaction prediction performance. We address this point in ***Evaluation of predicted interactions and complexes***:

“While all interactomes shared some interactions with the CORUM and PANTHER sets (Figure S6A-B), the extent of this overlap did not drive predictive performance for these external interaction sets (Figure S6C-D).”

Supplemental Figure 6. Overlap with external complex and pathway interaction sets and quality of predicted complexes. A-B) Distribution of the fraction of CORUM (A) and PANTHER (B) interactions present in each interactome. Interactomes with the highest overlap are labeled. C-D) Correlation between interaction prediction performance (mean of MPS and L3 results) and interaction overlap with CORUM (C) and PANTHER (D), with associated Pearson's correlation value.

- Figure 5D: I don't get what is an "orthogonal" network coverage?

In Figure 5D, we use the corpus of interactomes to assess the support for predicted interactions, such that predicted interactions from one interactome are assessed based on how frequently they appear in other interactomes. We have clarified the legend for **Figure 5D**:

“D) Distribution of network coverage for the top 100 interactions predicted by each interactome. A network coverage of 0 indicates that an interaction was not reported in any of the 45 interactomes. The top bar plot shows the number of edges in the corresponding interactome.”

And have expanded the corresponding **Methods: Interaction Prediction - Prediction of external interaction sets:**

“Finally, we defined the set of interactomes as an additional external set. We took the top 100 predictions from analysis with the full interactomes and assessed the support for these interactions based on their coverage within the corpus of other interactomes. Those not appearing in any of the 45 interactomes (network coverage of zero) were considered previously unreported.”

Methods

- protein "abundance", as used in the main text, should be preferred over "protein expression", which is used in the method section.

We have updated all occurrences.

- For protein abundance, it is not clear if the data are provided in categorical levels or quantitative values or both as categories are mentioned after the sentences about mean computation.

We have clarified in ***Methods: Data Collection and Processing - Collation and processing of gene metadata*** as follows:

“The categorical abundance levels provided by HPA were converted to numerical values (Not detected=0, Low=1, Medium=2, High=3). We then consolidated these numerical values for each gene by taking the mean of associated protein abundance values across all tissues.”

Significance

This manuscript is a follow-up on a previous paper (Huang et al. 2018 Cell Systems) with extended interactome and benchmarks. Molecular networks are widely used directly to study gene and protein cellular functions as well as data features in a large palette of machine learning computational biology tools. But the number of available networks has been growing steadily and it's often not easy for the users to navigate the interactome zoo and decide which network to consider. In this context, the work proposed here is expected to be very helpful to an audience of computational biologists using network data. The manuscript is clear and well-written, organised in logical steps. The figures are informative and of high quality.

We appreciate the reviewer's positive assessment of our work, and we feel the comments given have improved the clarity and impact of our manuscript.

Reviewer #2

Evidence, reproducibility and clarity

The manuscript submitted by Wright et al describe a comparative analysis of 46 human protein-protein interaction (PPI) networks that were generated from different sources being experimental, curation, or composite. The PPI networks were evaluated based on two criteria: their performance for the prediction of disease genes and for the prediction of novel PPIs. A couple more analyses were presented relating to network coverage, incorporation of networks in other resources and GO enrichments. The authors conclude that larger composite networks are better for disease gene prediction while smaller signaling pathway-related networks are better for PPI prediction.

While the presented analyses are mostly thorough the focus on disease gene and PPI prediction seem very restrictive given the rather general statements made in the title and abstract that these networks were evaluated for their ability to generate biological insights. Other relevant applications are for example (physiological) gene function predictions using guilt-by-association or the prediction of protein complex assemblies.

We thank the reviewer for appreciating our thorough analysis. To address this comment, we have significantly broadened the gene physiological function prediction and prediction of protein complex assemblies in **new Supplemental Figure 4C-F** and **new Supplemental Figure 6E-H**. We find that these analyses are complementary to the prediction of disease function and binary complexes already performed.

We present the gene function prediction in ***Gaps remain in coverage of the human proteome***:

“To further test each interactome's ability to represent GO annotations, we tested physiological gene function prediction using guilt-by-association⁴⁰. HumanNet⁴¹, Wan¹⁰, Havugimana⁴², DIP²⁸, Reactome⁴³, and PTMCode2⁴⁴ showed the best recapitulation of GO

annotations (Figure S4C), with performance generally best for cellular compartments (Figure S4D-F). However, it should be noted that some resources, such as HumanNet and Havugimana, utilize GO to prioritize network interactions during construction^{41,42}.”

And the protein complex prediction in ***Evaluation of predicted interactions and complexes:***

“In addition to predicting binary interactions, analysis of interactome structures can also be used to predict functional assemblies of genes and proteins. We constructed hierarchical representations of each interactome by applying the Hierarchical community Decoding Framework (HiDeF⁵⁵). Networks such as ConsensusPathDB⁴⁸, FunCoup⁵¹, PROPER¹⁷, TFLink⁵⁶ and hu.MAP⁵⁷ identified a large number of small assemblies, while others, such as STRING⁴⁷, HumanNet⁴¹, and Pathway Commons⁵⁸, tended to identify larger assemblies (Figure S6E). Interactomes such as DIP²⁸ and Havugimana⁴² captured the greatest number of CORUM complexes (Figure S6F), while PrePPI⁵⁹ and SIGNOR³⁹ identified complexes with the highest GO similarities among complex members (Figure S6G). It should be noted that some interactomes make use of CORUM or GO during construction, leading to potential biases. Therefore, we also examined the mean clustering coefficient of predicted complexes, observing the highest clustering for GeneMANIA⁵⁰, STRING⁴⁷, and HumanNet⁴¹ (Figure S6H). Reactome⁴³ and ReactomeFI¹⁴ were consistently among the top-performing interactomes across all three metrics.”

Supplemental Figure 4. C-F) Gene Ontology (GO) gene function prediction via neighbor-voting, with the area under the precision-recall curve (AUPRC) calculated using 5-fold cross-validation with error bars indicating 95% confidence intervals. B) Mean AUPRC across all GO annotations compared to mean Null AUPRC calculated from gene degree alone. D-F) Mean AUPRC for GO annotations within each GO branch (BP: biological process, CC: cellular component, MF: molecular function). Purple dashed lines show the mean null AUPRC from node degree alone across all interactomes.

Supplemental Figure 6. E) Size distributions of protein assemblies detected via hierarchical community detection. F-H) Evaluation statistics for the 15 top-performing networks based on assemblies with less than 200 proteins. F) Number of CORUM complexes recovered, defined as a Jaccard similarity of > 0.5 between any predicted assembly and a CORUM complex. G) Mean functional coherency of predicted assemblies based on the mean semantic similarity of GO annotations between assembly proteins (Methods). H) Mean clustering coefficient of predicted assemblies.

The accompanying methods can be found in Methods: Representation Analysis - Gene function prediction via guilt-by-association:

“Guilt-by-association analysis was performed using Extending Guilt by Association by Degree⁴⁰ (EGAD), as implemented in the R package EGAD v1.32.0. This method utilizes a neighbor-voting framework and compares the results to a baseline prediction based solely on the degree of genes within the network. We identified a broad group of GO annotation gene sets from GO associations previously downloaded from NCBI. Again, all genes associated with a term were assumed to be associated with all super terms. Analysis was performed for all GO sets with at most 250 genes. We performed the guilt-by-association analysis using 5-fold cross-validation, holding out one-fifth of genes in each GO set for each fold, followed by calculation of the AUPRC for prediction of the held-out genes. Additional AUPRC values were reported for each fold using predictions based on genes ranked by node degree, acting as a measure of degree bias⁴⁰. For each interactome, the observed AUPRC values were averaged over all GO terms with a minimum of 10 genes and a maximum of 250 genes present in the interactome”

Please see also Methods: Interaction and Complex Prediction - Prediction and evaluation of hierarchical assemblies:

“Hierarchical community structures were generated using the Hierarchical community Decoder Framework⁵⁵ (HiDeF) via the Python package `hidef` v1.1.5. All interactomes were assessed with a maximum resolution of 10, and all other parameters default. A recovered CORUM complex was defined as one reaching a Jaccard similarity of at least 0.5 with any predicted assembly. The GO Score was calculated based on the semantic similarity of genes present within each assembly⁸⁹. The pairwise semantic similarity, implemented in `goatools`⁷⁶, is calculated separately for each branch of GO for each pair u, v of genes within the assembly N according to Equation 12.

$$GO(N) = \frac{1}{3} \sum_{u \in N} \sum_{v \in N, v > u} BP(u, v) + MF(u, v) + CC(u, v) \quad (\text{Equation 12})$$

The mean clustering coefficient was calculated for all assemblies using the `average_clustering` function from `NetworkX`. GO Scores and clustering coefficients were calculated for all assemblies with ≤ 200 genes.”

The conclusions lack a more detailed discussion, e.g. why is it that the signaling networks perform better for PPI prediction when evaluated against predictability with AlphaFold? Could there be a potential bias because these smaller signaling networks like SIGNOR, SPIKE and REACTOME appear to be high quality networks due to manual curation.

We have expanded our **Discussion** to address two possible reasons why signaling networks perform well in prediction of PPIs:

“In particular, the top-performing networks included many with improved size-adjusted performance in gene set recovery, indicating the quality rather than quantity of interactions is crucial for interaction prediction. Our *in silico* assessment with AlphaFold-Multimer further reinforced the importance of network selection, with a small subset of interactomes such as SIGNOR³⁹ and Reactome⁴³ predicting interactions with high AlphaFold-Multimer support (Figure 5F, Table 1). While experimental physical protein-protein interaction networks such as Havugimana⁴² remain optimal for predicting stable multiplex complexes (Figure 5B), our results suggest that networks incorporating signaling and pathway interactions could be more frequently utilized to predict binary protein interactions. The majority of previously unreported interactions supported by our AlphaFold-Multimer analysis involved receptor and regulatory interactions, suggesting that *in silico* approaches may indeed help fill gaps in interactomes caused by experimental limitations related to transmembrane proteins and dynamic interactions (Figure 2D).”

I guess the authors have ensured that predicted PPIs do not have a resolved structure in the PDB yet? I wonder to which extent potential circularity in literature curation on one side and training on structures in the PDB on the other side lead to higher validation rates by AF for these types of networks. Would these signaling networks consist more of direct PPIs compared to other networks? This could also lead to higher AF modelling successes. Just to give some ideas.

To address this comment, we have assessed the presence of experimentally resolved structures from PDB across all AlphaFold-Multimer analyses (**new Supplemental Figure 7A-C,E**). These results have been added to ***In silico assessment of predicted interactions***:

“Applying AlphaFold-Multimer (AF) to random selections of interactions with varying network coverage, we observed significantly higher interface predicted template modeling (ipTM) scores for interactions present in more than 25 interactomes (Figure 5E), especially for the most supported interactions (≥ 33 supporting interactomes). This pattern was consistent with higher proportions of experimentally resolved structures amongst highly supported interactions (Figure S7A). We further identified nine interactomes containing interactions with significantly higher ipTM scores than randomly generated protein pairs (Figure 5F, FDR < 0.15). Again, we observed a correlation between the presence of experimentally resolved structures in PDB and ipTM scores, although these generally accounted for fewer than 20% of tested interactions (Figure S7B-C)...

The AF status of an [previously unreported] interaction did not depend on whether experimentally resolved structures were available for its component proteins ($p = 0.12$, χ^2 -squared test), with all AF-supported interactions involving at least one protein with no structure in PDB (Figure S7E).”

Accompanying methods are in ***Methods: Data Collection and Processing - Collation and processing of gene metadata***:

“Lists of experimentally resolved structures in the Protein Data Bank (PDB)⁶¹ were downloaded on August 14, 2024. Interactions with experimentally resolved structures were broadly defined as any protein pair reported as part of a human complex with at least two distinct protein entities that could be mapped to NCBI Gene IDs via UniProt Accession Codes. Proteins with individual experimentally resolved structures were broadly defined as any protein identified from structures containing one distinct protein entity (including homodimers and partial structures) that could be mapped to NCBI Gene IDs via UniProt Accession Codes.”

Supplemental Figure 7. AlphaFold-Multimer predictions and presence of experimentally resolved structures from PDB. A) Distribution of experimentally resolved structures in PDB as a function of network coverage of interactions. B) Distribution of AlphaFold-Multimer ipTM scores and corresponding distribution of experimentally resolved structures in PDB for 50 randomly selected interactions from each interactome. Colored violins indicate the network has a higher mean ipTM score than randomly generated protein pairs. C) Correlations between the fraction of experimentally resolved structures in PDB and the mean ipTM score per network. D) ipTM scores for previously unreported interactions predicted by each interactome using the MPS algorithm. The green line represents the 95th percentile of ipTM scores for randomly generated protein pairs, with protein pairs above this threshold defined as AF-supported interactions. E) The distribution of experimentally resolved structures in PDB among previously unreported interactions classified as AF-supported and AF-unsupported. See also Supplemental Tables 4 & 5.

In the same way, a more in depth discussion and potential analysis might be warranted for why larger networks perform better for the prediction of disease genes. From the methods section I understand that disease genes were filtered to those being contained in a respective network. Did the authors thoroughly check for any other imbalance that might arise because of large differences in network sizes and how this impacts the gene sets used to evaluate the networks?

Indeed, we hypothesize that one reason larger networks perform better in gene set recovery is due to greater coverage of the disease gene sets. When filtering gene sets to those contained in a respective network, differences in the size of the gene set and the particular genes included inevitably arise, as few genes are present across all networks. To address the effect of these gene set differences, we have performed a secondary analysis where all parameters and gene sets are held constant, which we present in **new Supplemental Figure 5B-C**:

Supplemental Figure 5. B-C) Gene set recovery performance for 16 interactomes, controlling for gene set size. Given a set of disease-associated genes, the ‘original’ set represents the genes present in each interactome independently, while the ‘reduced’ set represents the maximal subset of genes present in all 16 interactomes. B) Mean gene set recovery performance with literature and genetic gene sets for each interactome relative to interactome size, measured as the number of interactions. The line shows a log-linear fit with a 95% confidence interval. C) Distribution of regression slopes between interactome size and gene set recovery for literature and genetic gene sets, as measured for size-adjusted performance. Arrows indicate the median regression slope.

We comment on this result in the section entitled: ***Large composite networks remain the most powerful for disease gene prioritization***:

“When controlling for gene set coverage, a positive correlation was maintained between performance and network size (Figure S5B). However, the magnitude of this effect is significantly reduced ($p_{\text{Literature}} = 2.5 \times 10^{-36}$, $p_{\text{Genetic}} = 4.1 \times 10^{-16}$; Figure S5C), indicating that larger networks are, in part, advantaged by capturing a greater proportion of the known genes.”

The updated methods for this analysis are presented in the new section ***Methods: Gene Set Recovery Evaluation - Gene set coverage and interactome size***:

“We defined a subset of interactomes and gene sets to assess the influence of gene set coverage on the observed correlation between interactome size and gene set recovery performance. A representative set of 16 interactomes (BioPlex 293T, BioGRID, ConsensusPathDB, DIP, FunCoup, GIANT, HPRD, HumanNet, InBio, InnateDB, MatrixDB, PTMCode2, PrePPI, ReactomeFI, Signalink, STRING) was chosen to represent the range

of interactome sizes. From the previously analyzed literature and genetic gene sets ('original sets'), we identified the maximal subsets ('reduced sets') such that all members of the maximal subset were present in all 16 interactomes. We then required that the maximal subsets contain a minimum of 30 genes for literature sets ($n = 343$) or 15 genes for genetic sets ($n = 46$). Using these reduced sets, we assessed the gene set recovery performance for the 16 interactomes using the procedure outlined above with the mean optimal gene set recovery parameters of $p = 0.3$ and $\alpha = 0.64$. To avoid incorrect labeling of false positives, all other genes from the original sets not present in the reduced sets were excluded from the calculation of the performance metrics. For matched original and reduced gene sets, we fit a linear regression between the mean performance score and the \log_{10} interaction count. We further calculated the size-adjusted performance and extracted the slopes of the fitted models for each gene set. We compared the slopes between the original and reduced sets via a Wilcoxon paired test, as implemented in `scipy`⁸¹."

The use of AlphaFold for its ability to predict structural models of PPIs is a reasonable complementary approach to gain confidence in PPI predictions but is overstated in its ability to almost replace experimental validation. The use validation in this context is over-optimistic. To the best of my understanding, no clear performance evaluations exist of AF for its ability to predict PPIs. I suggest more carefully rephrasing the corresponding results section.

We thank the reviewer for acknowledging the utility of AlphaFold for the *in silico* investigation of PPI interactions. It was not our intention to position AlphaFold as a replacement for experimental validation. We have renamed the corresponding results section to ***In silico assessment of predicted interactions*** and updated the text to remove use of the term 'validated':

"Though not a substitute for experimental validation, AlphaFold-Multimer²¹ provides an *in silico* approach for assessing potential physical protein interactions...

We identified 115 interactions with an ipTM score in the 95th percentile of scores from randomly generated protein pairs (ipTM > 0.56), which we defined as AF-supported interactions."

Given different performances of the networks for PPI prediction as evaluated by AF, it is also rather unclear what the specific recommendations of the authors are with respect to which networks should be used for PPI prediction.

We now provide explicit recommendations in **new Table 1**, as described below under the Significance section of this review. We have also added further comments to the **Discussion**:

"In particular, the top-performing networks included many with improved size-adjusted performance in gene set recovery, indicating the quality rather than quantity of interactions is crucial for interaction prediction. Our *in silico* assessment with AlphaFold-Multimer further reinforced the importance of network selection, with a small subset of interactomes such as SIGNOR³⁹ and Reactome⁴³ predicting interactions with high AlphaFold-Multimer support (Figure 5F, Table 1). While experimental physical protein-protein interaction networks such as Havugimana⁴² remain optimal for predicting stable multiplex complexes (Figure 5B), our results suggest that networks incorporating signaling and pathway interactions could be more frequently utilized to predict binary protein interactions."

The methods section could provide at places some more detail, i.e. how do the PPI prediction algorithms L3 and MPS(T) work? What is the similarity by which proteins are assessed for the MPS?

We have expanded the section **Methods: Interaction and Complex Prediction - Interaction prediction algorithms** as follows:

“The L3 algorithm¹³ ranks predicted interactions based on the number of paths of length three between two nodes, normalized by the degree of the intermediate nodes. Formally, it calculates a degree normalized probability of interaction based on the number of paths of length three connecting two nodes X and Y such that:

$$L3_{XY} = \sum_{U,V} \frac{a_{XU}a_{UV}a_{VY}}{\sqrt{k_U k_V}} \quad (\text{Equation 8})$$

Where k_U is the degree of node U and $a_{XU} = 1$ if proteins X and U interact and zero otherwise. All possible interactions are ranked based on decreasing $L3$ scores to determine the most likely predicted interactions.

The Maximum similarity and Preferential attachment Score algorithm⁵² (MPS) utilizes two measures of topology to rank potential interactions based on the hypothesis that the probability of two proteins interacting is proportional to the similarity between one protein and the most similar interactors of the other protein. The maximum topological similarity between two nodes is calculated from the similarities of their neighboring nodes in the network. First, we define $N(u)$ as the set of direct neighbors of node u and the Jaccard Similarity between two nodes u and v based on the similarity of their neighbors:

$$J(u, v) = \frac{|N(u) \cap N(v)|}{|N(u) \cup N(v)|} \quad (\text{Equation 9})$$

The maximum topological similarity between nodes x and y is then defined based on the maximum Jaccard similarity between any neighbor of x : $a \in N(x)$ and y , and the maximum Jaccard similarity between any neighbor of y : $b \in N(y)$ and x such that:

$$\text{Max}_{sim}(x, y) = \max_{a \in N(x)} J(a, y) + \max_{b \in N(y)} J(b, x) \quad (\text{Equation 10})$$

The preferential attachment score P for two nodes x and y is defined as:

$$P_{xy} = k_x \times k_y \quad (\text{Equation 11})$$

Where k is the node degree. All possible interactions were ranked based on both similarity metrics, consistent with previous implementations²⁰.”

It is also not clear what the authors mean by random protein pairs. At places this could mean a random selection of PPIs from a network, at other places this could mean a generation of random pairs of proteins. This should be better specified.

We have clarified this language throughout the **Results** and **Methods** to refer to ‘samples of network protein pairs’ and ‘randomly generated protein pairs.’

The results section could be improved in clarity if clear hypotheses were formulated motivating the choice for the analyses presented.

The main analyses in this study aim to provide an unbiased and hypothesis-free examination of available network resources. However, some follow-up analyses were motivated by clear hypotheses, and we have now stated these more explicitly.

In **Gaps remain in coverage of the human proteome**:

“We next hypothesized that differences and biases in the genes represented in each interactome would correspond to gaps in biological function.”

In ***Not all interaction types are created equal***:

“The highest-performing networks were composite networks comprising many evidence types, leading us to hypothesize that different interaction types did not contribute equally to the observed performance.”

Figure 2F is referred to as Figure 2I. Please check the results and discussion section.

The paragraph below, which refers specifically to the node-level GO analysis in Figure 2F, was mistakenly omitted from ***Gaps remain in coverage of the human proteome*** in the submitted manuscript file. We thank the reviewer for catching this oversight and have now included it. Please note that the original Figure 2F is now Figure 2D. We have also checked all figure references.

“We next hypothesized that differences and biases in the genes represented in each interactome would correspond to gaps in biological function. Gene set enrichment analysis using Gene Ontology (GO) annotations showed that experimental interactomes favor translational machinery while being systematically under-enriched for receptor-ligand and cyclase activities, possibly due to experimental limitations (Figure 2D). Furthermore, many curated and small composite networks such as SIGNOR³⁹, PID v2.0³⁵, and DIP²⁸ were enriched for inflammation and immune response genes, while protein glycosylation and transporter activity were generally underrepresented.”

Significance

In general, it remains unclear how the findings reported in the study advance the field in understanding which of the networks are recommended for the use of which application.

To address this comment, we have constructed an explicit table of recommendations (**new Table 1**), which provides an overview of the best-performing networks in each analysis as well as important considerations for network users. We reference this table throughout the ***Discussion***, for example:

“Our analysis highlights the leading interactomes across various applications (Table 1), revealing significant differences depending on the task at hand.”

Table1. Summary of interactome recommendations across network biology applications.

Outcome/Analysis		Top Network Choices	Important Considerations
Best Gene Coverage	Experimental	BioPlex293T , BioPlex HCT116 , HuRI	 • Many databases continue to expand • Experimental and curated interactomes contain high-quality interactions but are more incomplete • Composite interactomes depend on a wide variety of sources
	Curated	BioGRID , HINT , HPRD	
	Composite	HIPPIE , GeneMANIA , Pathway Commons	
Biological Representation	Least Citation and Expression Bias	GeneMANIA , STRING , HumanNet	 • Bias towards highly cited and highly expressed genes reflects a combination of experimental and curation biases, and gene functional importance • Networks should be examined for coverage of relevant tissues and biological functions
	Highest Citation Bias	PID2 , DIP , PhosphoSitePlus	
	Highest Expression and Abundance Bias	Havugimana , ProteomeHD , Wan	
	Gene Function Prediction	HumanNet , Havugimana , Reactome (GO:BP), PTMCode2 (GO:CC), ReactomeFI (GO:MF)	
Best Disease Gene Prioritization	Literature	HumanNet , ConsensusPathDB , PCNet2.0	 • Large networks demonstrate higher performance using network propagation due to capturing more genes of interest and a greater range of prior knowledge • Certain interaction types, such as co-citation and domain similarity, contribute strongly to performance
	Genetic	STRING , HumanNet , FunCoup , PCNet2.0	
	Excluding co-citation interactions	FunCoup , PCNet2.2 , STRING (CC-free)	
Best Interaction Prediction	Self-interactions	DIP , ReactomeFI , Reactome , ProteomeHD	 • Smaller networks with high-quality and relevant interaction types perform better • Interaction prediction with large networks can become computationally intractable • Some interactomes incorporate CORUM and PANTHER
	CORUM interactions	Havugimana , PTMCode2 , ReactomeFI	
	PANTHER interactions	SIGNOR , Reactome , HPRD	
Best Protein Assembly Prediction	CORUM Recapitulation	DIP , Havugimana , Reactome	 • Different network topologies lead to different hierarchical assembly structures • Some interactomes incorporate CORUM and GO during network construction
	Functional Coherence	PrePPI , SIGNOR , TFLink	
Best in silico assessment (AlphaFold)	Self-interactions	SIGNOR , PID2 , HPRD , ProteomeHD	 • Combining interaction prediction with in silico assessment may help address systematic gaps in interactomes • Experimental validation remains necessary for confirming predicted interactions
	Previously unreported interactions	SIGNOR , SPIKE , Reactome	

Biases in the networks arising from literature curation, expression levels, and gene conservation are not clearly quantified but this would be important to understand which networks should be used for which application, also to avoid circularity for some applications.

We agree with the reviewer that looking at global trends of these biases does not give a complete picture of potential network biases. We have replaced the original Figures 2C-E with **new main Figure 2C**, which quantifies expression, abundance, and citation count per interactome.

New Figure 2C. C) Median citation, mRNA expression, and protein abundance were calculated for protein-coding genes in each interactome and compared to the median of all protein-coding genes. Colored bars show networks with significantly higher median values than a baseline of all protein-coding genes (permutation test, $q < 0.05$, Bonferroni-corrected).

The original Figures 2C-E, and an additional figure examining gene conservation can be found in **updated Supplemental Figure 1**. Further, we have expanded the analysis to include tissue-specific gene expression and protein abundance (**new Supplemental Figures 2 & 3**), which we feel will be particularly informative for selecting networks for specific disease indications.

We present these results in **Gaps remain in coverage of the human proteome**:

“Looking at the gene citation counts for protein-coding genes, we found that genes in PID v2.0³⁵, DIP²⁸, and PhosphoSitePlus³⁶ showed the most skew towards high citation counts (Figure 2C)...

Among the interactomes, we observed that experimental networks tended to show bias towards highly expressed genes and abundant proteins (Figure 2C). These same networks also tended to enrich for highly conserved genes (Figure S1F) and demonstrated under-enrichment for tissue-specific genes (Figure S2, S3).”

The accompanying methods can be found in **Methods: Data Collection and Processing - Collation and processing of gene metadata**:

“Positional gene conservation scores (phyloP) were sourced from the UCSC Genome Browser⁷⁷ (<https://hgdownload.soe.ucsc.edu/goldenPath/hg38/phyloP30way/>), calculated using the PAST package for multiple alignment of 29 vertebrate species to the hg38 human genome. PhyloP scores from the reference chromosomes were aggregated for each transcript using BEDOPS v2.4.41⁷⁸ based on coding sequences (CDS) derived from GENCODE⁷⁹ v46 Basic Gene Annotation (reference chromosomes only, https://ftp.ebi.ac.uk/pub/databases/genocode/Gencode_human/release_46/genocode.v46.basic.annotation.gff3.gz). The final gene conservation score was defined per gene as the mean phyloP across all positions within the gene’s CDS.”

And **Methods: Representation Analysis - Gene-level annotations**:

“For each annotation (citation count, mRNA expression, protein abundance, and gene conservation), the median value was calculated for each interactome as the median value for protein-coding genes contained in the interactome. This value was compared to the median

value for all protein-coding genes via permutation test, whereby for an interactome with N genes, 10000 random samples of size N were taken from the set of all protein-coding genes. Empirical p-values were Bonferroni corrected...

Sets of tissue-enhanced genes for mRNA expression and protein abundance were defined using the criteria outlined by the Human Protein Atlas (HPA³⁷). Using mRNA Expression data from GTEx, all protein-coding genes were classified as one of a) Low Expression - TPM < 1 in all tissues; b) Tissue enriched - five-fold higher TPM in one tissue compared to all other tissues; c) Group enriched - five-fold higher TPM in 2-7 tissues compared to all other tissues; d) Tissue enhanced - five-fold higher TPM in one tissue compared to the average of all other tissues; or e) Broadly expressed - all genes not otherwise classified. The genes classified as Tissue enriched, Group enriched, or Tissue Enhanced for a given tissue were considered part of the tissue-enhanced mRNA expression gene set for that tissue. All protein-coding genes were separately classified based on the quantified HPA protein abundance levels of associated proteins (see Collation and processing of gene metadata). Low abundance was defined as abundance < 0.5, and Tissue enriched, Group enriched, and Tissue enhanced genes were defined based on threefold, rather than fivefold, higher abundance levels. All genes classified as Tissue enriched, Group enriched, or Tissue Enhanced for a given tissue were considered part of the tissue-enhanced protein abundance gene set for that tissue. The set of protein-coding genes in each interactome was tested for enrichment of tissue-enhanced genes using a Fisher's Exact Test."

New Supplemental Figure 1F. F) Median gene conservation scores for protein-coding genes in each interactome, compared to the median of all protein-coding genes. Colored bars show networks with significantly higher median conservation scores than the baseline of all protein-coding genes (permutation test, $q < 0.05$, Bonferroni-corrected).

Supplemental Figure 2. Enrichment for tissue-enhanced gene expression per interactome. Color represents the enrichment q-value (Fisher’s Exact test, BH-corrected), with blue showing under-enrichment and red showing over-enrichment. The size of the points represents the effect size. Tissue-enhanced gene sets were defined from GTEx to include all genes classified as “Tissue Enriched,” “Group Enriched,” and “Tissue Enhanced” based on the HPA criteria (Methods). “Low Expression” genes are defined as those with a maximum TPM < 1 across all networks, and all genes not otherwise classified are considered “Broadly Expressed.”

Supplemental Figure 3. Enrichment for tissue-enhanced protein abundance per interactome. Color represents the enrichment q-value (Fisher’s Exact test, BH-corrected), with blue showing under-enrichment and red showing over-enrichment. The size of the points represents the effect size. Tissue-enhanced protein sets were defined to include all proteins classified as “Tissue Enriched,” “Group Enriched,” and “Tissue Enhanced” based on the HPA criteria. “Low Abundance” proteins were defined as tissues with mean abundance < 0.5, and all proteins not otherwise classified are considered “Broadly Abundant.”

This study would be of potential interest to a specialized group of computational researchers interested in the use of PPI networks for predicting biological function.

Our study is indeed intended for researchers interested in the use of molecular networks to understand gene function, a sizable community that includes a broad cross-section of ‘omics scientists. We appreciate the reviewer’s feedback and feel the comments given have improved and clarified our key findings.

21st Oct 2024

Manuscript Number: MSB-2024-12635-T

Title: State of the Interactomes: an evaluation of molecular networks for generating biological insights

Dear Dr. Ideker,

Thank you for the submission of your revised manuscript to Molecular Systems Biology. We have now received the enclosed reports from the referees that were asked to re-assess it. As you will see the reviewers are now globally supportive and I am pleased to inform you that we will be able to accept your manuscript pending the following final amendments:

- 1) Please download and fill out our "Author Checklist", which is published along with the published paper. You can find this checklist on our Author Guidelines page at: <https://www.embopress.org/page/journal/17444292/authorguide>
- 2) Please provide the manuscript in a .docx format, figures should be removed and uploaded separately, with no track changes.
- 3) In the main manuscript file, please include the corresponding author's email address on the title page, after the affiliations.
- 4) Please include keywords to max. 5.
- 5) Please rename the 'Availability of data and materials' section to 'Data availability' and format according to the example below:
"The datasets and computer code produced in this study are available in the following databases:
- Chip-Seq data: Gene Expression Omnibus GSE46748 (<https://www.ncbi.nlm.nih.gov/geo/query/acc.cgi?acc=GSE46748>)
- Modeling computer scripts: GitHub (<https://github.com/SysBioChalmers/GECKO/releases/tag/v1.0>)
- [data type]: [full name of the resource] [accession number/identifier] ([doi or URL or identifiers.org/DATABASE:ACCESSION])"
- 6) Please rename "Declarations of Interest" to "Disclosure and competing interests statement". Please also add the following disclaimer: "Trey Ideker is a member of the Advisory Editorial Board of The EMBO Journal. This has no bearing on the editorial consideration of this article for publication." We updated our journal's competing interests policy in January 2022 and request authors to consider both actual and perceived competing interests. Please review the policy <https://www.embopress.org/competing-interests> and update your competing interests if necessary.
- 7) Author contributions: Please remove it from the manuscript and specify author contributions in our submission system. CRediT has replaced the traditional author contributions section because it offers a systematic machine-readable author contributions format that allows for more effective research assessment. You are encouraged to use the free text boxes beneath each contributing author's name to add specific details on the author's contribution. More information is available in our guide to authors:
<https://www.embopress.org/page/journal/17574684/authorguide#authorshipguidelines>
- 8) References: Please correct the reference citation in the reference list. Where there are more than 10 authors on a paper, please list the first 10, followed by "et al.". Please check "Author Guidelines" for more information.
<https://www.embopress.org/page/journal/17574684/authorguide#referencesformat>
- 9) Our journal encourages inclusion of *data citations in the reference list* to directly cite datasets that were re-used and obtained from public databases. Data citations in the article text are distinct from normal bibliographical citations and should directly link to the database records from which the data can be accessed. In the main text, data citations are formatted as follows: "Data ref: Smith et al, 2001" or "Data ref: NCBI Sequence Read Archive PRJNA342805, 2017". In the Reference list, data citations must be labeled with "[DATASET]". A data reference must provide the database name, accession number/identifiers and a resolvable link to the landing page from which the data can be accessed at the end of the reference. Further instructions are available at .
- 10) Data not shown: We do not allow statements/conclusions with "data not shown". Please explain which outliers are not shown and why in the legend of Supplemental Figure 1.
- 11) In the Methods, please take care of the following:
- 12) All Materials and Methods need to be described in the main text using our 'Structured Methods' format. According to this format, the Methods section includes a Reagents and Tools Table (listing key reagents, experimental models, software and relevant equipment and including their sources and relevant identifiers) followed by a Methods and Protocols section describing the methods, ideally using a step-by-step protocol format. The aim is to facilitate adoption of the methodologies across labs. Please download and fill our Reagents and Tools Table template (.docx), which you can find in our author guidelines:
<https://www.embopress.org/page/journal/17444292/authorguide>.
When submitting your revised manuscript, please do not include the Reagents and Tools Table in the Methods section of the manuscript but upload it as a separate file choosing the file type "Reagent Table".
An example of a Method paper with Structured Methods can be found here:
<https://www.embopress.org/doi/10.15252/msb.20178071>. "
- 13) Please place individual sections of the manuscript in the following order: Title page - Abstract & Keywords - Introduction - Results - Discussion - Methods - Data Availability - Acknowledgements - Disclosure and Competing Interests Statement - References - Figure Legends - Expanded View Figure Legends.
- 14) For the figures and figure legends, please take care of the following:
- Please remove all figures from main manuscript file and leave only main figure legends placed after the references. Main figures should be uploaded as individual, high-resolution files. Regarding the supplementary figures, you can upload up to 5 as Expanded View (EV) Figures (EV figures will be displayed in the main HTML of the paper in a collapsible format). EV figures

need to be uploaded as separate Figure files as well, with their legends in the manuscript file, after the main figure legends, and renamed with the following nomenclature 'Figure EV1'. The remaining figures should be compiled in one PDF file labeled "Appendix" with their legends. Please ensure that the figure legends are included with the figures in the appendix, and that the appendix has a table of contents with page numbers. Please also ensure that the nomenclature for the figures and tables is correct, i.e. "Appendix Figure S1" and "Appendix Table S1". Please check "Author Guidelines" for more information: <https://www.embopress.org/page/journal/17574684/authorguide#figureformat>

- There is a callout for a 'Table S1' in the manuscript - does this refer to Supplemental Table 1?

- Supplemental tables S2-S5 and S7-S8 should be renamed to Dataset EVx and source file names, titles, legends and ms callouts all need to be updated to Dataset EV1-EV#; legends should be removed from manuscript file and uploaded as a separate tab/sheet in each Excel file.

- Supplemental tables S1 and S6 should be renamed to Appendix Table S1-S2 and compiled in the Appendix PDF, also source file names, titles, legends and callouts in the manuscript all need to be updated to Appendix Table S1-S2 and the legends should be removed from ms file and uploaded above each table; References should be corrected to alphabetical with the first 10 authors being shown followed by et al.

- Please note that the box plots need to be defined in terms of minima, maxima, centre, bounds of box and whiskers, and percentile in the legends of figures 3d; 5e-g.

- Please note that information related to n is missing in the legends of figures 3d; 4b-c; 5e-g.

15) Synopsis:

- Synopsis image: Please provide a graphic that summarises the main findings of the manuscript on a glance and upload it as a high-resolution jpeg file 550 pixels wide x (300-600) pixels high.

- Synopsis text: Please provide a short standfirst (maximum of 300 characters, including space), limit the bullet points to max. 5 and upload it as a separate .doc file. Please write the bullet points to summarise the key NEW findings. They should be designed to be complementary to the abstract - i.e. not repeat the same text. We encourage inclusion of key acronyms and quantitative information (maximum of 30 words / bullet point). Please use the passive voice.

16) Source Data: Our colleague Hannah Sonntag will contact you about whether Source Data need to be provided for your manuscript. If so, please ensure that a completed Source Data checklist is uploaded (checklist will be sent to you by Hannah), along with a single source data file (zipped) per figure, with the panels clearly visible in the folder structure.

17) As part of the EMBO Publications transparent editorial process initiative (see our policy here:

https://www.embopress.org/transparent-process#Review_Process), Molecular Systems Biology will publish online a Peer Review File (PRF) to accompany accepted manuscripts. This file will be published in conjunction with your paper and will include the anonymous referee reports, your point-by-point response and all pertinent correspondence relating to the manuscript. Let us know whether you agree with the publication of the PRF and as here, if you want to remove or not any figures from it prior to publication. Please note that the Authors checklist will be published at the end of the PRF.

18) Please provide a point-by-point letter INCLUDING my comments and your detailed responses (as Word file).

I look forward to reading a new revised version of your manuscript as soon as possible.

Yours sincerely,

Poonam Bheda, PhD
Scientific Editor
Molecular Systems Biology

Reviewer #1:

The authors have answered my reviewer comments to my full satisfaction. I especially appreciate the clarity with which responses have been provided. This very much facilitated the review process. I consider this manuscript suitable for publication in MSB. I spotted a small typo in figure S7C -> PBD should be PDB.

Reviewer #2:

The authors answered all my comments raised during the first submission of the manuscript.

Rev_Com_number: RC-2024-02501
New_manu_number: MSB-2024-12635-T
Corr_author: Ideker
Title: State of the Interactomes: an evaluation of molecular networks for generating biological insights

Manuscript Number: MSB-2024-12635-T

Corresponding Author: Trey Ideker

Point-by-point description of revisions for: State of the Interactomes: an evaluation of molecular networks for generating biological insights.

1) Please download and fill out our "Author Checklist", which is published along with the published paper. You can find this checklist on our Author Guidelines page at:
<https://www.embopress.org/page/journal/17444292/authorguide>

The completed Author Checklist has been uploaded.

- 2) Please provide the manuscript in a .docx format, figures should be removed and uploaded separately, with no track changes.
- 3) In the main manuscript file, please include the corresponding author's email address on the title page, after the affiliations.
- 4) Please include keywords to max. 5.

Points 2-4 have been addressed in the uploaded files.

5) Please rename the 'Availability of data and materials' section to 'Data availability' and format according to the example below:
"The datasets and computer code produced in this study are available in the following databases:
- Chip-Seq data: Gene Expression Omnibus GSE46748
(<https://www.ncbi.nlm.nih.gov/geo/query/acc.cgi?acc=GSE46748>)
- Modeling computer scripts: GitHub (<https://github.com/SysBioChalmers/GECKO/releases/tag/v1.0>)
- [data type]: [full name of the resource] [accession number/identifier] ([doi or URL or identifiers.org/DATABASE:ACCESSION)]"

This section now reads:

“Data Availability

Data presented in the figures are available in Datasets EV1-6, and full details of source databases utilized are provided in Appendix Table S1. Computer code and network resources produced in this study are available in the following databases:

- Computer scripts for data processing, analysis and visualization: GitHub (https://github.com/sarah-n-wright/Network_Evaluation_Tools/releases/tag/v0.2.1)
- All networks utilized in this study: NDEX (<https://www.ndexbio.org/index.html#/user/bae4da70-e22d-11ee-9621-005056ae23aa>)
 - Standardized source networks: NDEX (<https://doi.org/10.18119/N95C9J>)
 - PCNet2.0 network: NDEX (<https://doi.org/10.18119/N9JP5J>)
 - PCNet2.1 network: NDEX (<https://doi.org/10.18119/N9DW40>)
 - PCNet2.2 network: NDEX (<https://doi.org/10.18119/N9960N>)

6) Please rename "Declarations of Interest" to "Disclosure and competing interests statement". Please also add the following disclaimer: "Trey Ideker is a member of the Advisory Editorial Board of The EMBO Journal. This has no bearing on the editorial consideration of this article for publication."

We updated our journal's competing interests policy in January 2022 and request authors to consider both actual and perceived competing interests. Please review the policy <https://www.embopress.org/competing-interests> and update your competing interests if necessary.

This section now reads:

“Disclosure and competing interests statement

TI is a co-founder, member of the advisory board, and has an equity interest in Data4Cure and Serinus Biosciences. TI is a consultant for and has an equity interest in Ideaya Biosciences. The terms of these arrangements have been reviewed and approved by the University of California San Diego in accordance with its conflict-of-interest policies. TI is a member of the Advisory Editorial Board of The EMBO Journal. This has no bearing on the editorial consideration of this article for publication.”

7) Author contributions: Please remove it from the manuscript and specify author contributions in our submission system. CRediT has replaced the traditional author contributions section because it offers a systematic machine-readable author contributions format that allows for more effective research assessment. You are encouraged to use the free text boxes beneath each contributing author's name to add specific details on the author's contribution. More information is available in our guide to authors:

<https://www.embopress.org/page/journal/17574684/authorguide#authorshipguidelines>

This section has been removed and contributions added during submission.

8) References: Please correct the reference citation in the reference list. Where there are more than 10 authors on a paper, please list the first 10, followed by "et al.". Please check "Author Guidelines" for more information.

<https://www.embopress.org/page/journal/17574684/authorguide#referencesformat>

All citations and reference lists have been updated to comply with EMBO guidelines.

9) Our journal encourages inclusion of *data citations in the reference list* to directly cite datasets that were re-used and obtained from public databases. Data citations in the article text are distinct from normal bibliographical citations and should directly link to the database records from which the data can be accessed. In the main text, data citations are formatted as follows: "Data ref: Smith et al, 2001" or "Data ref: NCBI Sequence Read Archive PRJNA342805, 2017". In the Reference list, data citations must be labeled with "[DATASET]". A data reference must provide the database name, accession number/identifiers and a resolvable link to the landing page from which the data can be accessed at the end of the reference. Further instructions are available at

<https://www.embopress.org/page/journal/17574684/authorguide#referencesformat>.

We have added several data citations to better link to key datasets used for the analysis of interactomes.

10) Data not shown: We do not allow statements/conclusions with "data not shown". Please explain which outliers are not shown and why in the legend of Supplemental Figure 1.

All box plots in Figure S1 (now Figure EV1) have been updated to show all points lying outside the whisker boundaries of the box plots.

11) In the Methods, please take care of the following:

12) All Materials and Methods need to be described in the main text using our 'Structured Methods' format. According to this format, the Methods section includes a Reagents and Tools Table (listing key reagents, experimental models, software and relevant equipment and including their sources and relevant identifiers) followed by a Methods and Protocols section describing the methods, ideally using a step-by-step protocol format. The aim is to facilitate adoption of the methodologies across labs.

Please download and fill our Reagents and Tools Table template (.docx), which you can find in our author guidelines: <https://www.embopress.org/page/journal/17444292/authorguide>.

<https://www.embopress.org/doi/10.15252/msb.20178071>. "

We have completed the Reagents and Tools Table and uploaded it as a separate file, and have also converted key analysis methods into step-by-step protocols and highlighted important processing steps in bullet point format.

13) Please place individual sections of the manuscript in the following order: Title page - Abstract & Keywords - Introduction - Results - Discussion - Methods - Data Availability - Acknowledgements - Disclosure and Competing Interests Statement - References - Figure Legends - Expanded View Figure Legends.

All sections have been placed in the correct order.

14) For the figures and figure legends, please take care of the following:

- Please remove all figures from main manuscript file and leave only main figure legends placed after the references. Main figures should be uploaded as individual, high-resolution files. Regarding the supplementary figures, you can upload up to 5 as Expanded View (EV) Figures (EV figures will be displayed in the main HTML of the paper in a collapsible format). EV figures need to be uploaded as separate Figure files as well, with their legends in the manuscript file, after the main figure legends, and renamed with the following nomenclature 'Figure EV1'. The remaining figures should be compiled in one PDF file labeled "Appendix" with their legends. Please ensure that the figure legends are included with the figures in the appendix, and that the appendix has a table of contents with page numbers. Please also ensure that the nomenclature for the figures and tables is correct, i.e. "Appendix Figure S1" and "Appendix Table S1". Please check "Author Guidelines" for more information: <https://www.embopress.org/page/journal/17574684/authorguide#figureformat>
- There is a callout for a 'Table S1' in the manuscript - does this refer to Supplemental Table 1?

- All figures have been removed from the manuscript file. Legends for main figures 1-5, and Expanded View Figures EV1-5 (formerly Figures S1,4,5,6&8) are included after the references.
- All main and EV figures have been uploaded as high-resolution .eps files.

- Supplemental Figures 2, 3, 7, & 9 are included in the Appendix PDF as Appendix Figures S1-4, along with their legends.
- The callout for 'Table S1' has been updated to 'Appendix Table S1'

- Supplemental tables S2-S5 and S7-S8 should be renamed to Dataset EVx and source file names, titles, legends and ms callouts all need to be updated to Dataset EV1-EV#; legends should be removed from manuscript file and uploaded as a separate tab/sheet in each Excel file.

- Supplemental tables S1 and S6 should be renamed to Appendix Table S1-S2 and compiled in the Appendix PDF, also source file names, titles, legends and callouts in the manuscript all need to be updated to Appendix Table S1-S2 and the legends should be removed from ms file and uploaded above each table; References should be corrected to alphabetical with the first 10 authors being shown followed by et al.

- Please note that the box plots need to be defined in terms of minima, maxima, centre, bounds of box and whiskers, and percentile in the legends of figures 3d; 5e-g.

- Please note that information related to n is missing in the legends of figures 3d; 4b-c; 5e-g.

- Supplemental Tables 1 & 6 are now included in the Appendix as Appendix Tables S1 & S2.
- Supplemental Tables S2-S5 and S7-S8 have been renamed Datasets EV1-6. Each dataset has been uploaded as a separate Excel file, with the names, titles, and legends included in a separate tab.
- All supplemental table references have been updated to comply with the EMBO style.
- We have added Appendix Table S3 to more formally cite the sources of experimental datasets.
- All legends for box and violin plots have been updated to fully define the centers, boxes, whiskers, and ranges.
- For all figures containing box plots, violin plots, or error bars, we have ensured n information is available in the figure and/or legend.

15) Synopsis:

- Synopsis image: Please provide a graphic that summarises the main findings of the manuscript on a glance and upload it as a high-resolution jpeg file 550 pixels wide x (300-600) pixels high.

- Synopsis text: Please provide a short standfirst (maximum of 300 characters, including space), limit the bullet points to max. 5 and upload it as a separate .doc file. Please write the bullet points to summarise the key NEW findings. They should be designed to be complementary to the abstract - i.e. not repeat the same text. We encourage inclusion of key acronyms and quantitative information (maximum of 30 words / bullet point). Please use the passive voice.

We have added the synopsis text and image and uploaded them in standalone files.

16) Source Data: Our colleague Hannah Sonntag will contact you about whether Source Data need to be provided for your manuscript. If so, please ensure that a completed Source Data checklist is uploaded (checklist will be sent to you by Hannah), along with a single source data file (zipped) per figure, with the panels clearly visible in the folder structure.

Per SourceData Scientific Coordinator: "Thank you for submitting all required source data files and thus making your research findable and accessible."

17) As part of the EMBO Publications transparent editorial process initiative (see our policy here: https://www.embopress.org/transparent-process#Review_Process), Molecular Systems Biology will publish online a Peer Review File (PRF) to accompany accepted manuscripts. This file will be published in conjunction with your paper and will include the anonymous referee reports, your point-by-point response and all pertinent correspondence relating to the manuscript. Let us know whether you agree with the publication of the PRF and as here, if you want to remove or not any figures from it prior to publication. Please note that the Authors checklist will be published at the end of the PRF.

We agree to the publication of the PRF, and do not request any figures be removed.

Reviewer #1:

The authors have answered my reviewer comments to my full satisfaction. I especially appreciate the clarity with which responses have been provided. This very much facilitated the review process. I consider this manuscript suitable for publication in MSB. I spotted a small typo in figure S7C -> PBD should be PDB.

We thank the reviewer for the positive assessment of our revisions, which we feel greatly improved the clarity of our manuscript. We have fixed the typo mentioned.

Reviewer #2:

The authors answered all my comments raised during the first submission of the manuscript.

We thank the reviewer for their time and feedback for improving this manuscript.

Rev_Com_number: RC-2024-02501

New_manu_number: MSB-2024-12635-T

Corr_author: Ideker

Title: State of the Interactomes: an evaluation of molecular networks for generating biological insights

11th Nov 2024

Manuscript number: MSB-2024-12635R

Title: State of the Interactomes: an evaluation of molecular networks for generating biological insights

Dear Dr. Ideker,

Congratulations on an excellent manuscript, I am pleased to inform you that your manuscript has been accepted for publication in Molecular Systems Biology. Thank you for your comprehensive response to referee concerns. It has been a pleasure to work with you to get your paper to the acceptance stage.

Yours sincerely,

Poonam Bheda, PhD
Scientific Editor
Molecular Systems Biology
